# Glance2Gaze: Efficient Vision-Language Models from Glance Fusion to Gaze Compression

**Juan Chen**[1,2][*] **Honglin Liu**[2]**, Yingying Ao**[2]**, Ting Zhang**[3][†] **Yan Huang**[1][†]
**Xudong Liu**[2]**, Biao Li**[2]**, Jintao Fang**[2]

[1]School of Computer Science and Engineering, South China University of Technology
[2]Meituan Inc.      [3]School of Artificial Intelligence, Beijing Normal University
`csjchen@mail.scut.edu.cn, tingzhang@bnu.edu.cn, aihuangy@scut.edu.cn`
`{liuhonglin03,aoyingying,liuxudong18,yaowenyuan,fangjintao}@meituan.com`

## Abstract

Vision-language models heavily rely on visual representations, yet ensuring its efficiency remains a critical challenge. Most existing approaches focus on reducing visual tokens either at the visual encoder phase or during the LLM decoder stage. Inspired by human visual cognition, where an initial global glance precedes focused attention on semantically salient regions, we introduce Glance2Gaze, a cognitively inspired framework that mimics the human two-stage attention process. The framework consists of two key components: the Glance Fusion module, which integrates multi-layer vision transformer features with text-aware attention to generate a semantically enriched global representation, and the Gaze Compression module, which utilizes a novel query-guided mechanism to selectively compress visual tokens based on their semantic relevance. Experimental results on widely adopted benchmarks demonstrate that Glance2Gaze outperforms existing methods, achieving superior performance with equal or lower computational cost. Furthermore, it generalizes well to high-resolution and video scenarios, showcasing robust and scalable efficiency improvements in VLMs.

## 1   Introduction

Large Vision-Language Models (VLMs) [1–6] have made significant advances in image captioning, visual question answering, and multimodal dialogue understanding driven by the rapid progress of Large Language Models (LLMs) [7–11]. A typical VLM architecture consists of three key components: (i) a pre-trained visual encoder responsible for extracting dense image representations; (ii) a cross-modal projector that aligns visual features with textual embeddings; and (iii) a language model backbone that processes the combined visual-textual tokens to generate task-specific outputs. Central to this pipeline are visual tokens, whose quality critically impacts model performance and efficiency, making them a core focus in VLM design.

Visual tokens are conventionally extracted from images using pre-trained vision backbones such as CLIP-ViT [12], EVA [13], or InternImage [14], which generate fixed-length sequences of visual embeddings for downstream multimodal tasks. Recent advancements in large VLMs have shown that increasing the number of visual tokens can significantly improve performance by capturing finer visual details [4, 5, 15, 16]. However, denser visual token representations pose substantial computational challenges due to the quadratic complexity of attention mechanisms in LLMs, where

---

[*]This work was done when Juan Chen was an intern at Meituan Inc.
[†]Corresponding authors.

39th Conference on Neural Information Processing Systems (NeurIPS 2025).

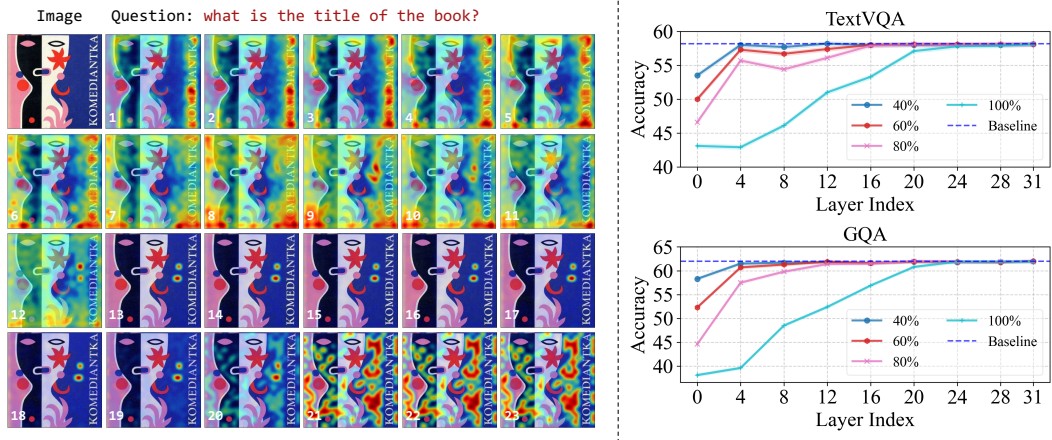

Figure 1: Left: Cross-modal attention heatmaps between instructions and visual tokens from different ViT layers, with numerical annotations indicating layer indices. Right: The pruning ratio-performance trade-off curve under LLaVA[3].

computational cost scales with the square of the total token count. Consequently, increasing token density exacerbates inference latency, memory consumption, and scalability issues.

In response to increasing complexity, substantial research has explored visual token selection, an approach that retains only the most informative tokens while discarding redundant or less relevant ones. This strategy is driven by the insight that not all visual tokens contribute equally to downstream tasks [17–23]. Existing methods can be broadly classified into two paradigms based on the stage at which pruning is applied. The first focuses on encoder-stage pruning [17, 19, 20, 22, 24], where token selection occurs within the visual encoder before fed into the language model. The second involves LLM-stage pruning [18, 21, 23, 25–27], where selection is performed alongside the language model decoder, often leveraging the full attention matrix to assess token importance. By selecting a subset of salient tokens, both paradigms aim to reduce token count while maintaining task performance, thereby enhancing computational efficiency. This paper also focus on efficient visual representation, providing novel insight from a cognitive perspective.

In this paper, we propose a novel framework for efficient vision-language models inspired by the glance-to-gaze mechanism observed in human visual cognition. Our work draws on the human eye movements that typically exhibit a two-stage pattern: initial fixations are broad and exploratory in nature, serving to rapidly acquire a global understanding of a scene, while subsequent fixations become increasingly focused [28]. Intuitively, human first perform a *glance* to capture the global layout and salient structures, followed by a *gaze* that zooms in on regions of interest for detailed inspection [29–32]. However, existing methods often aim to improve efficiency by selecting a subset of salient tokens, overlooking the hierarchical nature of human attention.

Motivated by this insight, we propose *Glance2Gaze*, a hierarchical visual token processing framework that emulates the dual-phase mechanism of rapid glancing followed by focused gazing. In contrast, our approach begins with a lightweight global scan to identify candidate regions and then progressively refines attention toward semantically rich subregions, enabling deeper cross-modal interaction. Specifically, Glance2Gaze comprises two complementary modules: Glance Fusion and Gaze Compression. Glance Fusion enables fast and holistic perception by dynamically aggregating multi-layer features rather than solely the penultimate ViT layer used in conventional approaches. We employ a text-aware attention to integrate hierarchical representations, enriching global semantics without increasing token count. As shown in Figure 1 (left), attention heatmaps across ViT layers indicate that each layer contributes distinctively to cross-modal understanding, highlighting the benefit of hierarchical feature aggregation for global context modeling. Building on the initial global understanding, Gaze Compression shifts the focus toward localized visual cues by progressively condensing visual tokens within the language model. This module is informed by the observation that token redundancy varies with decoding depth: shallow decoder layers are sensitive to pruning, whereas deeper layers remain robust under significant token reduction, as shown in Figure 1 (right).

We introduce a novel query-guided mechanism to selectively compress visual tokens based on their semantic relevance, facilitating a smooth transition from global to fine-grained visual reasoning.

We conduct extensive experiments to evaluate the effectiveness and efficiency of our proposed framework. Empirical results demonstrate that it consistently outperforms state-of-the-art baselines on both image and video understanding tasks, while maintaining comparable computational efficiency on the LLaVA backbone series. To ensure a comprehensive evaluation, we include a detailed analysis of computational cost, including inference latency comparison. Beyond empirical performance, our framework is grounded in cognitive principles drawn from human visual perception. By aligning model behavior with established cognitive mechanisms, our approach provides a principled, biologically inspired pathway toward more efficient vision-language models.

In summary, our key contributions are:

- We introduce a *Glance2Gaze*, a cognitively motivated two-stage visual token processing framework that mirrors human eye movement patterns, initial global glancing followed by focused gazing, to enhance efficiency in vision-language models.
- Building on the insight, we design a *Glance Fusion* module that aggregates multi-layer ViT features using text-aware attention, enriching global semantic understanding without increasing visual token count and a *Gaze Compression* module, an efficient iterative visual token compression method that uses a shared query pool to condense visual tokens, emulating the visual system's gaze process.
- We present an in-depth analysis about each component of the proposed framework. Empirically we demonstrate our approach delivers superior performance comparing with state-of-the-art baselines on both image understanding and video understanding tasks.

## 2 Related Work

**Vision-Language models.** Recent advances in LLMs like GPT [7] and LLaMA [9] have driven the emergence of VLMs [1–6, 33]. VLMs aim to process and understand information across multiple data modalities, including text, images, videos. LLaVA [2] pioneered open-source VLMs by introducing a visual-language instruction dataset and establishing a robust framework through the integration of CLIP-ViT-L-336px [12] visual encoder with the Vicuna [8] language model, thereby laying the groundwork for future architectures. However, it resizes each image into fixed number of visual tokens, hindering its performance in complex tasks. To enhance visual tokens, several methods [1, 4, 10, 15] have increased the number of image tokens to enhance model detail comprehension and reduce hallucinations. LLaVA-NeXT [15] dynamically divides images into patches, allowing the visual encoder to process them independently before concatenating them to form the visual representation. Meanwhile, Qwen2-VL [5] introduces dynamic resolution to accommodate high-resolution inputs and trains the visual encoder from scratch using extensive image-text pairs, resulting in improved performance on fine-grained tasks. Furthermore, video-based VLMs [34, 35] extract multiple frames to enhance video understanding, resulting in a substantial increase in token count. While increasing visual tokens aids in managing complex scenarios, the excessive volume can significantly hinder the practical application of VLMs in real-world settings. Recently, GG-Transformer [36] adopts a glance-and-gaze concept to improve the modeling of long-range dependencies and local contexts within ViTs. However, its objective differs fundamentally from ours, as it focuses on enhancing unimodal representational capacity in ViT, whereas our goal is to improve efficiency in VLM.

**Visual compression for VLMs.** Recent methods have concentrated on reducing the visual token count while aiming to preserve model performance. Prior works have explored compressing visual tokens before interfacing with the LLM [17, 19, 20, 22, 24, 33, 37, 38]. Resampler [39] utilizes a learnable query embedding to compress visual tokens from the vision encoder, while MQT [38] introduces random selection of queries to adapt token count to task needs. VisionZip [22] employs a training-free pruning strategy using encoder attention to identify and remove redundant tokens, similar to LLaVA-Prumerge [20]. These methods face challenges in optimizing performance and efficiency; excessive token retention leads to redundancy, whereas aggressive compression compromises accuracy. Another type of approaches [18, 21, 23, 25, 27] reduce token count in LLM decoders using attention matrices. FastV [18] proposes halving visual tokens post-second LLM layer, while SparseVLM [23] employs a training-free method to prune tokens based on text relevance scores. These rely on full attention matrices, making them incompatible with FlashAttn [40]. PDrop [21] attempts to address this by

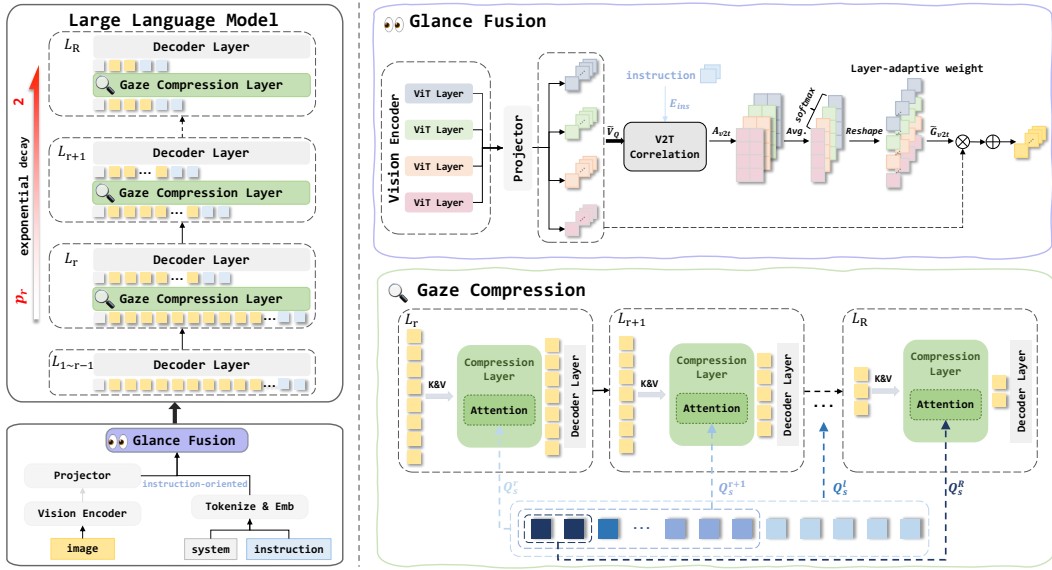

Figure 2: Diagram of the proposed Glance2Gaze framework. Glance Fusion combines hierarchical ViT features, guided by instruction, to produce enriched text-aware visual representations. Gaze Compression iteratively reduces visual tokens using a shared query embedding pool $Q_s$.

introducing a lightweight similarity calculation before attention. However, these methods miss the human-like glance-to-gaze transition, which highlights the importance of capturing a global visual context before compression—a crucial aspect often neglected.

Unlike these methods, the proposed Glance2Gaze framework is inspired by cognitive principles and consists of two steps: Glance and Gaze, to efficiently compress visual tokens. The Glance step enhances overall image perception by providing additional detail, while the Gaze step gradually focuses on more relevant local areas with a novel query-guided compression.

## 3 Method

The Glance2Gaze architecture is depicted in Figure 2. Our design places the Glance Fusion module before the LLM and embeds the Gaze Compression module within it. Glance Fusion enhances text-aware visual representations by integrating ViT layers prior to token compression, while Gaze Compression iteratively reduces visual tokens starting at a specific decoder layer, transitioning focus from global scenes to local regions.

**Revisiting VLMs.** In VLMs, an image input $I \in \mathbb{R}^{H \times W \times 3}$ is divided into $N$ discrete tokens via patch convolution and refined through $K$ transformer layers, yielding visual tokens $V \in \mathbb{R}^{K \times N \times d_v}$, where $d_v$ denotes the dimension of visual tokens. Typically, tokens from the penultimate layer are selected for input to the projector and subsequently the LLM. $R$ denotes the total number of decoder layers in the LLM.

### 3.1 Glance Fusion

Building on the preceding analysis, we introduce Glance Fusion to enhance global context understanding. While prior studies [41–43] explored improving visual features using pre-trained visual encoders, these methods overlook the crucial interplay between textual and visual signals, thus limiting their performance gains. Dense Connector [41] divides ViT layers into groups, averages tokens within each group, and concatenates them across channels. MMFuser [42] utilizes deep ViT features as queries to retrieve missing details from shallow features. In contrast, Glance Fusion employs text-conditioned attention to dynamically integrates ViT's multi-level features, enabling task-aware feature weighting without incurring significant computational overhead or added token count.

**Visual-instruction correlation.** We first partition the ViT into distinct hierarchical stages to mitigate computational overhead while preserving representational diversity. We strategically sample intermediate layers from ViT. Let $\mathbb{L} = \{l_1, l_2, ..., l_S\}$ denote the predefined set of layer indices (e.g., $l_s \in [1, 24]$ for a 24-layer ViT), where $S = |\mathbb{L}|$ specifies the number of selected layers. From the full-layer image tokens $\boldsymbol{V} \in \mathbb{R}^{K \times N \times d_v}$, we extract tokens corresponding to $\mathbb{L}$, yielding a lightweight hierarchical representation $\tilde{\boldsymbol{V}} \in \mathbb{R}^{S \times N \times d_v}$. These tokens are then projected into the LLM's embedding space via the projector $\mathcal{P}_v$, yielding $\tilde{\boldsymbol{V}}_Q$:

$$\tilde{\boldsymbol{V}} = concat(\boldsymbol{V}_{l_s}), l_s \in \mathbb{L}, \tag{1}$$

$$\tilde{\boldsymbol{V}}_Q = \mathcal{P}_v(\tilde{\boldsymbol{V}}) \in \mathbb{R}^{S \times N \times d_t}, \tag{2}$$

where $d_t$ denotes the dimension of text tokens.

Let $T_{ins}$ denotes the input instruction tokens, we first generate its textual embedding $\boldsymbol{E}_{ins} \in \mathbb{R}^{M \times d_t}$ using the LLM's native embedding layer, ensuring parameter-sharing consistency with linguistic processing, $M$ equals to the number of instruction tokens. To adaptively align textual semantics with hierarchical visual features, we introduce layer-specific projection layer $\{\mathcal{P}_t^s\}_{s=1}^S$, where $\mathcal{P}_t^s$ transforms $\boldsymbol{E}_{ins}$ into a subspace tailored for the $\tilde{\boldsymbol{V}}_Q^s$. Formally:

$$\boldsymbol{E}_{ins}^s = \mathcal{P}_t^s(\boldsymbol{E}_{ins}) \in \mathbb{R}^{M \times d_t}, s = 1, 2, ..., S. \tag{3}$$

For each layer $l_s$, the projected text embedding $\boldsymbol{E}_{ins}^s$ is paired with its corresponding ViT feature $\tilde{\boldsymbol{V}}_Q^s$ to compute cross-modal correlation score, enabling granularity-aware fusion of visual semantics. Specifically, we take $\tilde{\boldsymbol{V}}_Q^s$ as query and $\boldsymbol{E}_{ins}^s$ as key to compute scaled dot-product correlation score, resulting in an attention matrix $\boldsymbol{A}_{v2t}^s \in \mathbb{R}^{N \times M}$. We calculate the row-wise average to form the vector $\boldsymbol{g}_{v2t}^s \in \mathbb{R}^N$, which represents the correlation of each image token in the $l_s$-th ViT layer with all instruction tokens. Then we concatenate all $\boldsymbol{g}_{v2t}^s$ to form $\boldsymbol{G}_{v2t} \in \mathbb{R}^{S \times N}$:

$$\boldsymbol{A}_{v2t}^s = \frac{\tilde{\boldsymbol{V}}_Q^s \boldsymbol{E}_{ins}^{s\top}}{\sqrt{d_t}} \in \mathbb{R}^{N \times M}, s = 1, 2, ..., S, \tag{4}$$

$$\boldsymbol{g}_{v2t}^s = avg(\boldsymbol{A}_{v2t}^s) \in \mathbb{R}^N, s = 1, 2, ..., S, \tag{5}$$

$$\boldsymbol{G}_{v2t} = concat(\boldsymbol{g}_{v2t}^s) \in \mathbb{R}^{S \times N}. \tag{6}$$

**Instruction-oriented integration.** To optimally extract visual features that enhance instruction comprehension, we apply Softmax to $\boldsymbol{G}_{v2t}$ along the column dimension, yielding normalized correlation scores for each visual token across $S$ different layers. These scores act as dynamic weights to produce task-enhanced visual tokens $\boldsymbol{V}_Q$, resulting in:

$$\tilde{\boldsymbol{G}}_{v2t} = softmax(\boldsymbol{G}_{v2t}, dim = 0), \tag{7}$$

$$\boldsymbol{V}_Q = \sum_{s=1}^S \tilde{\boldsymbol{V}}_Q^s \odot \tilde{\boldsymbol{G}}_{v2t}^s \in \mathbb{R}^{N \times d_t}, \tag{8}$$

where $\odot$ denotes element-wise multiplication broadcasted along the token dimension. This method employs the correlation between instructions and ViT layers as weights to fuse their outputs, yielding task-enhanced visual features. These enhanced features, $\boldsymbol{V}_Q$, are then concatenated with text tokens for processing by the LLM.

### 3.2 Gaze Compression

We propose Gaze Compression, a parameter-efficient mechanism that iteratively reduces visual token count across decoder layers via a shared learnable query pool. Building on previous observation that shallow decoder layers are sensitive to pruning, we retain all visual tokens in the shallow decoder layers to fully understand them. Therefore, we decide to initiate the Gaze Compression from a predefined layer $r$. By retaining the full token count at shallow layers, we enable the LLM to thoroughly understand visual tokens and enhance overall comprehension, thus laying a solid foundation for subsequent compression.

**Progressive gaze compression.** To achieve token reduction, we define a monotonically decreasing sequence $P = [p_r, p_{r+1}, \ldots, p_R]$ with $p_r \geq p_{r+1} \geq \cdots \geq p_R$, where $p_l$ ($r \leq l \leq R$) represents the number of compressed visual tokens at the $l$-th decoder layer, which is less than the number of initial

visual tokens, namely $p_r < N$. Starting from layer $r$, a query-based compression attention operation compresses visual tokens before they are inputted to each decoder layer. At each $l$-th layer in the LLM, a learnable query $\boldsymbol{Q}_l \in \mathbb{R}^{p_l \times d_t}$ is initialized, with visual tokens from $l$-1-th layer $\boldsymbol{H}_{l-1}$ serving as key and value for cross-attention computation. The attention output yields compressed visual tokens for the decoder layer's processing. However, distinct queries across layers increase parameter count and complicate optimization for discrete visual token extraction. To mitigate this, a learnable query embedding $\boldsymbol{Q}_s \in \mathbb{R}^{p_r \times d_t}$ is shared across all pertinent decoder layers. Consequently, at the $l$-th layer, the first $p_l$ queries $\boldsymbol{Q}_s^l \in \mathbb{R}^{p_l \times d_t}$ are drawn from $\boldsymbol{Q}_s$, and visual tokens are compressed as follows:

$$\boldsymbol{H}_l = \mathcal{F}_o(softmax(\frac{\mathcal{F}_q(PE(\boldsymbol{Q}_s^l))\mathcal{F}_k(\boldsymbol{H}_{l-1})^\top}{\sqrt{d_t}})\mathcal{F}_v(\boldsymbol{H}_{l-1})), \tag{9}$$

where $\mathcal{F}_{o/q/k/v}$ denotes four distinct linear projection layers, and $PE$ refers to the 2D Rotary Positional Embedding [10] applied to $\boldsymbol{Q}_s^l$ for spatial information perception. The resulting output $\boldsymbol{H}_l \in \mathbb{R}^{p_l \times d_t}$ replaces the input $\boldsymbol{H}_{l-1}$, yielding compressed visual tokens. By continually compressing beyond layer $r$, visual tokens are reduced, lowering computational demands in the decoder layers. This progressive reduction mimics the fine gazing process, focusing visual tokens on critical image regions.

### 3.3 Efficiency Analysis

Following prior work [21, 22], we report the FLOPs of the image token component. Glance Fusion accounts for less than 2.72% of the total FLOPs, which is therefore negligible compared to the computational cost incurred by visual token processing within LLM. We thus provide additional details in the supplementary material and here exclude it from further analysis.

Consequently, we focus solely on the FLOPs associated with the LLM's processing of visual tokens, evaluating efficiency from two perspectives: 1) the FLOPs required to process visual token sequences through decoder layers, and 2) the overhead introduced by Gaze Compression, as outlined in Eq. 9.

For Vicuna [8] (featuring 3-layer MLP following attention), the FLOPs per layer is $4n_l d_t^2 + 2n_l^2 d_t + 3n_l d_t d_m$, where $n_l$ denotes the number of visual tokens ($n_l = N$ before layer $r$, $n_l = p_l$ thereafter) at $l$-th decoder layer and $d_m$ is the hidden dimension of FFN. The compression overhead in Eq. 9 includes attention computation and four linear layers, resulting in $2(p_l d_t^2 + p_{l-1} d_t^2 + p_l p_{l-1} d_t)$ FLOPs, assuming $p_{l-1} = N$ when $l = r$. Then the overall computational cost is:

$$C = \sum_{l=1}^{R} 4n_l d_t^2 + 2n_l^2 d_t + 3n_l d_t d_m + \sum_{l=r}^{R} 2(p_l d_t^2 + p_{l-1} d_t^2 + p_l p_{l-1} d_t). \tag{10}$$

For instance, by setting $r$ to 9 and using an exponentially decaying sequence $P = [256, \dots, 2]$ in LLaVA-1.5-7B, we maintain only 33.2% of the FLOPs while minimally affecting performance across several benchmarks. To manage different compression ratios, $p_R$ is always set to 2, with adjustments made to $r$ and $p_r$. Additional details are provided in the supplementary material.

## 4 Experiments

### 4.1 Image Understanding Tasks

**Benchmarks.** To validate the effectiveness of our method in image understanding tasks, we conducted experiments on ten mainstream datasets, including TextVQA [44], POPE [45], GQA [46], VQAv2 [47], SEEDBench [48], MMBench [49], MME [50], ScienceQA-IMG [51], MMVet [52] and LLaVA-Bench-in-the-wild [2].

**Implementation details.** We applied the proposed method to both LLaVA-1.5-7B [3] and the high-resolution backbone, LLaVA-NeXT-7B [15], which increases the input image resolution to 4x more pixels. For LLaVA, the vision encoder was frozen while the remaining parameters were fine-tuned using the LLaVA-665k [3] dataset, adhering to the original training settings. For LLaVA-NeXT, all parameters were unfrozen during fine-tuning. Given its proprietary code and training data, we used the Open-LLaVA-NeXT [53], an open-source replication, for training, following PDrop [21]. In the

Table 1: The performance of Glance2Gaze at various compression configurations on LLaVA-1.5-7B, with FLOPs and the final column indicating the relative proportion of visual token computation and performance compared to the original model. **Bold** for best and underline for second-best performers.

| Method | FLOPs | POPE | SQA | MME | GQA | SEED | MMB | VQAv2 | TextVQA | MMVet | LLaVA-B | Avg. |
|---|---|---|---|---|---|---|---|---|---|---|---|---|
| LLaVA-1.5-7B | 100% | 85.9 | 69.5 | 1862 | 61.9 | 58.6 | 64.7 | 78.5 | 58.2 | 31.1 | 66.8 | 100% |
| FastV | 33% | 64.8 | 67.3 | 1612 | 52.7 | 57.1 | 61.2 | 67.1 | 52.5 | 27.7 | 49.4 | 87.5% |
| SparseVLM | 33% | 85.3 | 68.7 | 1787 | 59.5 | 58.7 | 64.1 | 75.6 | 57.8 | **33.1** | 66.1 | 99.0% |
| PDrop | 33% | 82.3 | 70.2 | 1766 | 57.1 | 54.7 | 63.2 | - | 56.1 | 30.5 | - | 96.2% |
| VisionZip | 33% | 84.9 | 68.2 | **1834** | 60.1 | 57.1 | 63.4 | 77.4 | **57.8** | 32.6 | **66.7** | 99.1% |
| Glance2Gaze | 33% | **85.5** | **70.4** | 1812 | **61.5** | **58.7** | **64.5** | **77.6** | 57.2 | 32.7 | 66.4 | **99.9%** |
| FastV | 22% | 59.6 | 60.2 | 1490 | 49.6 | 55.9 | 56.1 | 61.8 | 50.6 | 28.1 | 52 | 83.2% |
| SparseVLM | 22% | **85.0** | 68.6 | 1746 | 58.4 | **58.2** | 64.5 | 73.8 | 56.7 | 29.0 | 62.7 | 96.3% |
| PDrop | 22% | 82.3 | 69.9 | 1664 | 56.0 | 53.3 | 61.1 | - | 55.1 | 30.8 | - | 94.4% |
| VisionZip | 22% | 83.7 | 68.3 | **1823** | 58.9 | 55.8 | 62.6 | 76.6 | **57.0** | 32.9 | 64.8 | 97.9% |
| Glance2Gaze | 22% | 84.5 | **70.2** | 1794 | **59.3** | 56.6 | 63.4 | **77.4** | 56.8 | **33.1** | 65.4 | 98.7% |
| FastV | 11% | 48.0 | 51.1 | 1256 | 46.1 | 51.9 | 48.0 | 55.0 | 47.8 | 25.8 | 46.1 | 73.8% |
| SparseVLM | 11% | 77.5 | 69.8 | 1589 | 53.8 | 52.2 | 60.1 | 68.2 | 53.4 | 24.9 | 57.5 | 89.0% |
| PDrop | 11% | 55.9 | 69.2 | 1092 | 41.9 | 40.0 | 33.3 | - | 45.9 | 30.7 | - | 73.5% |
| VisionZip | 11% | 80.9 | 68.8 | **1756** | **57.0** | 53.4 | 61.5 | 74.2 | **56.0** | 30.2 | 63.6 | 94.9% |
| Glance2Gaze | 11% | **83.1** | 69.1 | 1722 | 56.9 | **53.9** | **61.7** | **74.9** | 55.7 | **31.4** | **64.1** | **95.6%** |

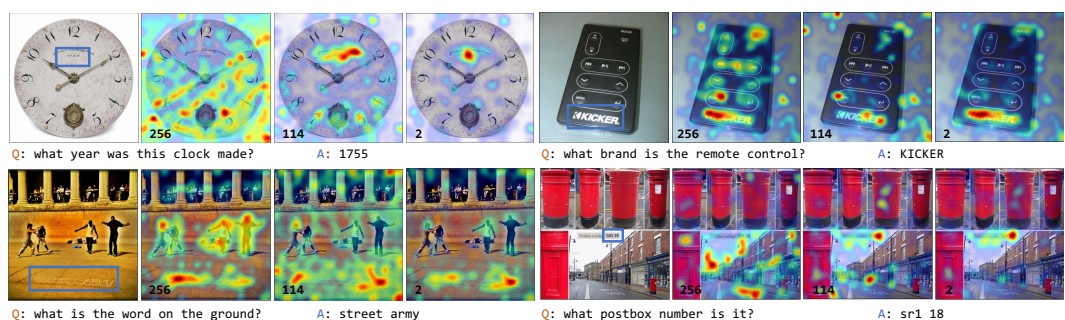

Figure 3: Visualization of different query embedding counts for compressing visual tokens from TextVQA [44]. Number within each image denotes $p_l$.

Glance Fusion module, $\mathbb{L}$ is set to {7, 13, 19, 23}. We implement the Gaze Compression strategy under different compression ratios. All experiments are conducted on 8 NVIDIA-A100-80G GPUs. Please refer to the supplementary material for more implementation details.

**Results on LLaVA.** Table 1 presents the performance of Glance2Gaze on LLaVA-1.5-7B [3], benchmarked against state-of-the-art approaches FastV [18], SparseVLM [23], PDrop [21], and VisionZip [22] across three FLOPs configurations, 33%, 22%, and 11%, indicating the computational ratio of visual processing compared to full token retention. Overall, Glance2Gaze consistently achieves top results at all compression levels and ranks as the best or second-best performer across nearly all datasets, showcasing its strong competitiveness. At 33% FLOPs, Glance2Gaze outperforms PDrop by 3.7% and VisionZip by 0.8%. Moreover, at 22% FLOPs, SparseVLM experience a 2.7% drop in accuracy, whereas Glance2Gaze only loses 1.2%, surpassing VisionZip by 0.8%. At the extreme compression ratio of 11%, Glance2Gaze retains 95.6% performance, exceeding VisionZip by 0.7% and SparseVLM by 6.6%, demonstrating its robust performance across various compression requirements.

**Results on LLaVA-NEXT.** LLaVA-NeXT extends LLaVA to handle high-resolution scenarios by dividing images into up to five patches, creating up to 2880 image tokens. Glance Fusion and Gaze Compression are applied independently to each sub-image. As shown in Table 2, Glance2Gaze exhibits greater advantages in high-resolution scenarios compared to standard resolutions. Glance2Gaze surpasses the second-best performer, VisionZip, with an average improvement of 1.2-1.3% across three FLOPs setting. Furthermore, Glance2Gaze surpasses VisionZip on nearly all datasets, demon-

Table 2: The performance of Glance2Gaze on LLaVA-NeXT-7B.

| Method | FLOPs | TextVQA | POPE | SQA | MME | GQA | MMB | VQAv2 | Avg. |
|---|---|---|---|---|---|---|---|---|---|
| LLaVA-NeXT-7B | 100% | 65.8 | 86.6 | 69.2 | 1801 | 63.8 | 67.2 | 80.5 | 100% |
| SparseVLM | 22% | 57.8 | - | 67.7 | 1772 | 60.3 | 65.7 | 77.1 | 95.4% |
| VisionZip | 22% | 60.8 | 87.6 | 67.9 | 1778 | 62.4 | 65.9 | 79.9 | 97.9% |
| Glance2Gaze | 22% | **62.1** | **89.2** | **69.9** | **1784** | **63.1** | **66.4** | **80.1** | **99.2%** |
| SparseVLM | 11% | 55.9 | - | 67.3 | 1694 | 57.7 | 64.3 | 73.4 | 92.3% |
| VisionZip | 11% | 59.3 | 86.2 | 67.5 | 1770 | 61.0 | **64.4** | 78.4 | 96.3% |
| Glance2Gaze | 11% | **60.5** | **87.5** | **69.9** | **1771** | **62.0** | 64.3 | **78.8** | **97.6%** |
| SparseVLM | 5% | 46.4 | - | 67.5 | 1542 | 51.2 | 63.1 | 66.3 | 85.0% |
| VisionZip | 5% | 57.3 | 83.4 | 67.5 | 1699 | 58.2 | 63.9 | 75.6 | 93.6% |
| Glance2Gaze | 5% | **58.6** | **84.4** | **68.8** | **1718** | **58.9** | **64.0** | **76.4** | **94.8%** |

Table 3: The performance of Glance2Gaze on Video-LLaVA at 6% FLOPs.

| Method | TGIF | MSVD | MSRVTT | ActivityNet | Avg. |
|---|---|---|---|---|---|
| Video-LLaVA | 47.1 | 69.8 | 56.7 | 43.1 | 100% |
| FastV | 23.1 | 38.0 | 19.3 | 30.6 | 52.1% |
| SparseVLM | 44.7 | **68.2** | 31.0 | 42.6 | 86.5% |
| VisionZip | 42.4 | 63.5 | 52.1 | 43.0 | 93.2% |
| Glance2Gaze | **45.0** | 66.5 | **52.4** | **43.8** | **96.2%** |

strating its robustness in preserving essential information during compression, especially in intricate OCR tasks such as TextVQA and VQAv2. The superior performance of Glance2Gaze in high-resolution scenarios likely stems from its ability to focus on spatially crucial details through the glance-to-gaze process, which becomes increasingly important as image complexity escalates.

## 4.2 Video Understanding Tasks

**Benchmarks.** Beyond image understanding, we applied the proposed method to video comprehension tasks, evaluating it on four widely adopted video-based question answering benchmarks: TGIF [54], MSVD [55], MSRVTT [55], and ActivityNet [56].

**Implementation details.** We use Video-LLaVA [34] to train a video VLM, which use Language Bind [57] as the vision encoder, extracts 8 frames from a video, encoding each into 256 visual tokens, culminating in 2048 image tokens in the end. To make a fair comparison with VisionZip and SparseVLM, $r$ and $p_r$ are set to 3 and 10 respectively to create a model with FLOPs reduced to 6% of the original. Please refer to supplementary material for more training details.

**Performance on Video-LLaVA.** In video understanding tasks, a substantial degree of frame-to-frame redundancy presents opportunities for computational efficiency without significantly compromising performance. Our method effectively capitalizes on this redundancy, achieving 96.2% of the performance using only 6% of FLOPs, whereas FastV suffers a 47.9% performance drop. Notably, Glance2Gaze outperforms VisionZip by 3% on average and ranks first on 3 out of 4 benchmarks, making it particularly suitable for real-time and large-scale video analysis applications.

## 4.3 Efficiency Comparison

Table 4 analyzes the computational costs, cuda time (i.e., the time required to generate the first token), overall latency (i.e., the total time to complete the entire sequence), and throughput on the TextVQA dataset, evaluated using a NVIDIA-A100-80G GPU, consistent across all compared baselines.

The results demonstrate that, under the same TFLOPs reduction ratio, Glance2Gaze not only delivers superior performance but also achieves the fastest inference speed, with a $6.39\times$ acceleration in prefilling time, a $3.15\times$ speedup in overall latency, and a $3.14\times$ improvement in throughput, effectively balancing efficiency and accuracy.

Table 4: Computational cost analysis comparison on LLaVA-NeXT-7B.

| Method | TFLOPs | Cuda Time (ms) | Latency (ms) | Throughput | TextVQA |
|--------|--------|----------------|--------------|------------|---------|
| LLaVA-NeXT | 53.83 | 313 | 475 | 10.884 | 65.8 |
| FastV | 2.76 | 170 (1.84×) | 358 (1.33×) | 12.272 (1.13×) | - |
| SparseVLM | 2.76 | 183 (1.71×) | 379 (1.25 ×) | 13.233 (1.22×) | 46.4 |
| VisionZip | 2.76 | 57 (5.49×) | 192 (2.47×) | 26.983 (2.48×) | 57.3 |
| Glance2Gaze | **2.67** | **49 (6.39×)** | **151 (3.15×)** | **34.121 (3.14×)** | **58.6** |

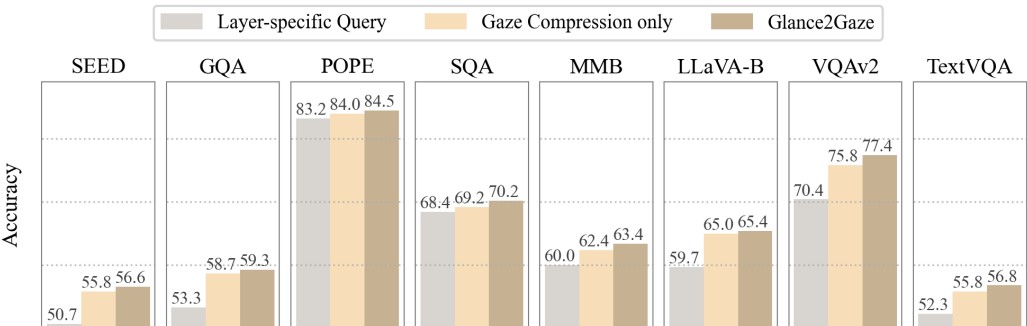

Figure 4: Ablation study evaluating the effect of a shared query pool in Gaze Compression comparing layer-specific query and the effect of Glance Fusion.

## 4.4 Analysis and Discussion

**Visualization of Gaze Compression.** Figure 3 illustrates the visual features learned by the query embedding at 33% FLOPs on TextVQA. Initially, with more visual tokens retained, the query captures broader global information. As compression progresses and fewer visual tokens are kept, focus shifts to regions containing the answer, demonstrating how the query learns to localize important areas through the gaze compression procedure, such as those involving text answers.

**The effect of Gaze Compression compared to attention-based strategies.** To rigorously assess the advantage of Gaze Compression over attention-based pruning, we evaluate both methods under an identical compression ratio. As shown in Table 5, attention-based pruning results in notable performance degradation. Even with subsequent fine-tuning, the baseline remains inferior to Glance2Gaze, underscoring that static pruning guided solely by local attention scores lacks the adaptability required for task-specific compression.

**The effect of a shared query pool.** We investigate the effects of using layer-specific queries in Gaze Compression, as shown in Figure 4. Using independent queries for each layer increases parameters and significantly damages performance across all datasets. This may be because the shared queries enhance pattern extraction from the same image, implicitly constraining the query's optimization domain. More ablations about Gaze Compression is available in the supplementary material.

**The effect of Glance Fusion.** Figure 4 demonstrates the impact of applying Gaze Compression alone and with Glance Fusion at 22% FLOPs on LLaVA-1.5-7B, resulting in substantial performance enhancements across all datasets. Notably, Glance Fusion improves accuracy by 1.0 and 1.6 points on TextVQA and VQAv2, respectively, where fine-grained image details are crucial for text understanding tasks. We emphasize that the additional computational overhead introduced by Glance Fusion is minimal, yet its effectiveness is substantial. More ablations about Glance Fusion is available in the supplementary material.

**Comparison with other fusion strategies.** We compare Glance Fusion with two representative fusion strategies, Dense Connector [41] and MMFuser [42], on LLaVA [3] with different scales, as summarized in Table 6. Dense Connector was not evaluated on LLaVA-1.5-13B and is therefore excluded from that setting. Glance Fusion consistently outperforms both baselines across model scales. On LLaVA-1.5-7B, it exceeds Dense Connector by 0.1–1.1 points, despite using fewer ViT layers (4 vs. 24), and surpasses MMFuser, which uses five layers, by 0.5–1.9 points. Similar

Table 5: Comparison of Gaze Compression with attention-based pruning methods.

| Method | POPE | SQA | MME | GQA | SEED | MMB | VQAv2 | TextVQA | MMVet | LLaVA-B | Avg. |
|---|---|---|---|---|---|---|---|---|---|---|---|
| w/o finetune | 83.4 | 69.1 | 1775 | 59.7 | 54.7 | 63.0 | 74.6 | 55.8 | 30.6 | 62.0 | 96.1% |
| w finetune | 84.7 | 69.1 | 1793 | 60.8 | 56.8 | 63.2 | 75.7 | 56.4 | 31.4 | 65.2 | 97.9% |
| Glance2Gaze | 85.5 | 70.4 | 1812 | 61.5 | 58.7 | 64.5 | 77.6 | 57.2 | 32.7 | 66.4 | 99.9% |

Table 6: Exploring the potential of Glance Fusion compared with other fusion strategies.

| Method | GQA | VQAv2 | SQA | TextVQA | POPE |
|---|---|---|---|---|---|
| LLaVA-7B | 62.0 | 78.5 | 66.8 | 58.2 | 85.9 |
| +Dense Connector | $63.8^{1.8\uparrow}$ | $79.5^{1.0\uparrow}$ | $69.5^{2.7\uparrow}$ | $59.2^{1.0\uparrow}$ | $86.6^{0.7\uparrow}$ |
| +MMFuser | $62.8^{0.8\uparrow}$ | $79.1^{0.6\uparrow}$ | $68.7^{1.9\uparrow}$ | $58.8^{0.6\uparrow}$ | $86.3^{0.4\uparrow}$ |
| +Glance Fusion | $64.1^{2.1\uparrow}$ | $79.6^{1.1\uparrow}$ | $70.6^{3.8\uparrow}$ | $59.4^{1.2\uparrow}$ | $87.2^{1.3\uparrow}$ |
| LLaVA-13B | 63.3 | 80.0 | 71.6 | 61.3 | 85.9 |
| +MMFuser | $63.4^{0.1\uparrow}$ | $80.1^{0.1\uparrow}$ | $71.2^{0.4\downarrow}$ | $59.9^{0.4\downarrow}$ | $87.5^{1.6\uparrow}$ |
| +Glance Fusion | $64.5^{1.2\uparrow}$ | $80.4^{0.4\uparrow}$ | $71.9^{0.3\uparrow}$ | $61.8^{0.5\uparrow}$ | $87.2^{1.3\uparrow}$ |

improvements are observed on LLaVA-1.5-13B. These results highlight Glance Fusion's superiority over task-agnostic fusion methods.

# 5  Conclusion and Limitation

This paper introduces a novel cognitive-inspired visual token compression through a two-stage *glance-to-gaze* approach. First, Glance Fusion dynamically merges multi-layer ViT features using a text-aware attention strategy to enhance image understanding. Next, Gaze Compression employs a query-based approach using a shared query pool to selectively compress visual tokens within LLM, mimicking the gaze process by progressively focusing on detailed local regions. Experimental results show that Glance2Gaze surpasses existing methods in performance with equal or reduced computational cost. While the overall framework is cognitively inspired, the internal mechanisms, particularly the fusion and compression strategies, remain largely heuristic and lack explicit modeling of cognitive processes. Future work could explore tighter integration with cognitive theories to further enhance both interpretability and performance.

## Acknowledgments

This work was supported by the National Key Research and Development Program of China under Grant 2024YFE0105400, the Guangdong Provincial Key Field R&D Program under Grant 2024B0101040008, and the Guangzhou Science and Technology Plan Project – Key R&D Plan under Grant 2024B01W0007. This work was jointly supported by the Fundamental Research Funds for the Central Universities under Grant 2243100004 and 2253500001.

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

# A  Supplementary Material

## A.1  Broader Impact

The proposed Glance2Gaze advances vision-language models by introducing a cognitively inspired framework that harmonizes efficiency and performance through its two-stage attention mechanism. By mimicking human visual cognition—first capturing global context and then focusing on salient details—the method significantly reduces computational overhead while maintaining or improving accuracy across diverse tasks. This innovation broadens accessibility to advanced visual-language systems, enabling deployment in resource-limited settings such as edge devices or real-time applications. Its scalability to high-resolution and video inputs further extends practical utility in fields like medical imaging, autonomous systems, and multimedia analysis. Environmentally, the reduced computational demand aligns with sustainable AI development by lowering energy consumption. As a generalizable paradigm, Glance2Gaze also inspires future research in biologically inspired attention mechanisms, fostering interdisciplinary advancements in efficient multimodal learning.

## A.2  Efficiency Analysis

Previous works [18, 21–23] typically measure efficiency by the computational cost of visual tokens in LLMs, a method we also adopt. Additionally, we analyze the extra computational load introduced by Glance Fusion.

**FLOPs within Glance Fusion.** Considering the computational cost incurred by Glance Fusion, we account for the FLOPs arising from all projection operations of both visual (Eq. 2) and textual (Eq. 3) inputs, as well as the overhead introduced by the visual-text correlation computation (Eq. 4). Specifically, the visual mapping introduces $SNd_vd_t$ FLOPs, while the text mapping contributes $SMd_t^2$ FLOPs. Furthermore, the visual-text correlation involves $S$ matrix multiplications, resulting in $SNMd_t$ FLOPs. Overall, the total FLOPs within the Glance Fusion module can be expressed as:

$$C_{glance} = SNd_vd_t + SMd_t^2 + SNMd_t. \tag{11}$$

**FLOPs within LLM (Visual Tokens).** We assess FLOPs of visual tokens within the LLM by examining two factors: (i) the standard forward pass through the decoder, and (ii) the additional cost introduced by the Gaze Compression module. For the visual token processing in Vicuna-7B [8], using a 3-layer MLP as FFN and 32 decoder layers, each layer incurs $4n_ld_t^2 + 2n_l^2d_t + 3n_ld_td_m$ FLOPs, where $n_l$ and $d_m$ denote the number of visual tokens and the hidden dimension of the FFN, respectively. In our framework, $n_l$ is 576 before layer $r$ and $p_l$ thereafter, with $l$ representing the LLM layer index. The FLOPs from Eq. 9 include 2 matrix multiplications and 4 linear projections, totaling $2p_{l-1}p_ld_t + 2p_{l-1}d_t^2 + 2p_ld_t^2$. Therefore, the total FLOPs within LLM are:

$$C_{llm} = \sum_{l=1}^{R} 4n_ld_t^2 + 2n_l^2d_t + 3n_ld_td_m + \sum_{l=r}^{R} 2(p_{l-1}p_ld_t + p_{l-1}d_t^2 + p_ld_t^2). \tag{12}$$

We found $C_{glance}$ to be negligible compared to $C_{llm}$. For instance, in LLaVA-1.5-7B, with $r$ set to 3 and $p_r$ to 100, $C_{llm}$ amounts to 1.1T FLOPs while $C_{glance}$ is only 0.03T FLOPs, yielding a proportion of 0.0272. Given its insignificance, $C_{glance}$ can be omitted from detailed computational analysis, allowing us to concentrate on the more impactful $C_{llm}$ for performance assessment.

## A.3  More Implementation Details

### A.3.1  Image Understanding Tasks

**Training Recipes.** Following LLaVA [3], we employ a two-stage training strategy for Glance2Gaze. In the first stage, we align image-text pairs by retaining the LLaVA architecture and training only the projector on the LLaVA-558K dataset for one epoch with a batch size of 256 and a learning rate of 1e-3. In the second stage, we incorporate Glance Fusion and Gaze Compression into LLaVA, training all parameters except the visual encoder for one epoch with a batch size of 128 and a learning rate of 2e-5.

To train LLaVA-NeXT-7B [15], we employ Open-LLaVA-NeXT [53], an open-source replication, due to proprietary restrictions on the original code and training sets. Our approach involves a two-stage

Table 7: Compression configurations employed to LLaVA-1.5-7B.

| FLOPs | $r$ | $P$ |
|---|---|---|
| 33% | 9 | [256, 209, 170, 139, 114, 93, 76, 62, 50, 41, 33, 27, 22, 18, 15, 12, 10, 8, 6, 5, 4, 3, 2, 2] |
| 22% | 7 | [121, 103, 88, 75, 64, 54, 46, 40, 34, 29, 24, 21, 18, 15, 13, 11, 9, 8, 7, 6, 5, 4, 3, 3, 2, 2] |
| 11% | 3 | [100, 87, 77, 67, 59, 52, 45, 40, 35, 30, 27, 23, 20, 18, 16, 14, 12, 10, 9, 8, 7, 6, 5, 4, 4, 3, 3, 2, 2, 2] |

Table 8: Compression configurations employed to LLaVA-NeXT-7B.

| FLOPs | $r$ | $P$ |
|---|---|---|
| 22% | 7 | [121, 103, 88, 75, 64, 54, 46, 40, 34, 29, 24, 21, 18, 15, 13, 11, 9, 8, 7, 6, 5, 4, 3, 3, 2, 2] |
| 11% | 3 | [100, 87, 77, 67, 59, 52, 45, 40, 35, 30, 27, 23, 20, 18, 16, 14, 12, 10, 9, 8, 7, 6, 5, 4, 4, 3, 3, 2, 2, 2] |
| 5% | 2 | [25, 23, 21, 19, 18, 16, 15, 14, 13, 12, 11, 10, 9, 8, 7, 7, 6, 6, 5, 5, 4, 4, 4, 3, 3, 3, 3, 2, 2, 2, 2] |

training process. Similar to LLaVA-1.5-7B, the proposed Glance2Gaze is integrated only in the second stage. In the first stage, we focus solely on training the projector with a batch size of 256 and a learning rate of 1e-3 for one epoch using the LLaVA-558K dataset. In the second stage, Glance2Gaze is embedded into the LLaVA-NeXT architecture, with all parameters including the visual encoder unfrozen for training over two epochs with a batch size of 128 and a learning rate of 2e-5, utilizing the same dataset as in [53].

**Implementation Details.** In Glance Fusion, $\mathbb{L}$ is consistently set to $\{7, 13, 19, 23\}$ to capture fine-detailed features. For Gaze Compression, we adjust $r$ and $p_r$ to manage different computation configurations, as detailed in Table 7 and 8.

LLaVA-NeXT processes high-resolution images by dividing them into multiple sub-images (up to 5) for individual handling by the visual encoder, which are subsequently concatenated within the LLM. Both Glance Fusion and Gaze Compression are applied to each sub-image independently.

### A.3.2 Video Understanding Tasks

**Training Recipes.** Following Video-LLaVA [34], we employ a two-stage training strategy. In the first stage, only the projector is trained for one epoch with a batch size of 256 and a learning rate of 1e-3, using the LLaVA-558K dataset and a subset of Valley [58], while retaining the original architecture. In the second stage, we integrate Glance Fusion and Gaze Compression into Video-LLaVA, freeze the vision encoder, and finetune all other parameters with a batch size of 128 and a learning rate of 2e-5, using the LLaVA-665K dataset and 100k video-text instructions from Video-ChatGPT [34].

**Implementation Details.** In Video-LLaVA [34], 8 frames are sampled from each video, with each frame encoded into 256 tokens, resulting in 2048 tokens per video. For Glance Fusion, $\mathbb{L}$ is set to $\{7, 13, 19, 23\}$ by default. Gaze Compression is applied to each frame, with $r = 3$ and $p_r = 10$, yielding a model that operates at 6% of the FLOPs required for full token retention.

### A.3.3 More Ablation Studies

**More ablation on the effect of a shared query pool.** Figure 5 presents additional ablation results comparing layer-specific query embeddings with a shared query pool in Gaze Compression. While the layer-specific approach increases parameters, it significantly degrades performance across all datasets.

**More ablation on the effect of Glance Fusion.** To assess the importance of global glance fusion prior to gaze compression, we compare Gaze Compression alone with the full Glance2Gaze in

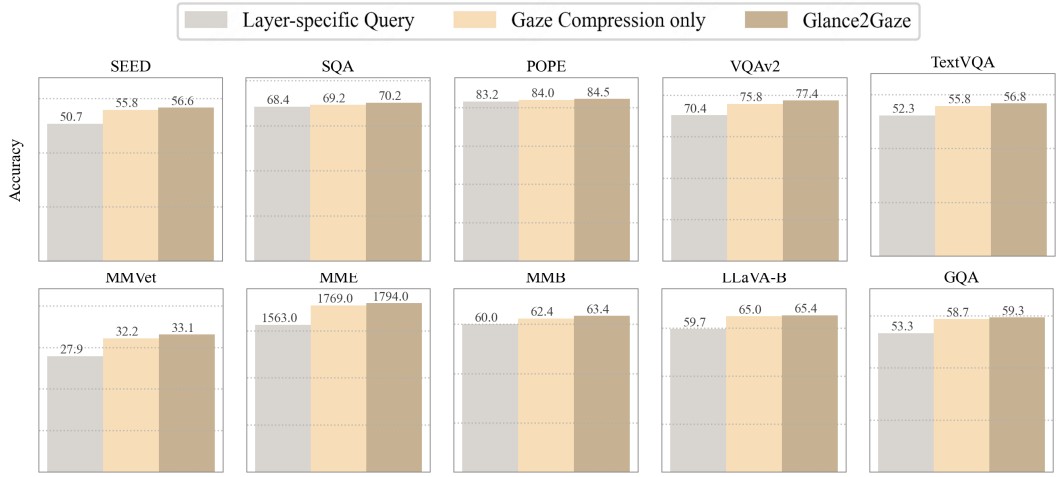

Figure 5: More ablation results on significance of shared query pool and the effect of Glance Fusion.

Table 9: Ablation studies on $r$ and $p_r$, evaluated on VQAv2 dataset.

| Method | $p_r$=256 | | | | | $r$=9 | | | |
|--------|-----------|------|------|------|------|-----------|------------|------------|------------|
| | $r$=1 | $r$=3 | $r$=9 | $r$=17 | $r$=25 | $p_r$=32 | $p_r$=128 | $p_r$=256 | $p_r$=529 |
| FLOPs | 12% | 17% | 33% | 56% | 78% | 24% | 28% | 33% | 42% |
| Acc | 70.2 | 74.6 | 77.6 | 78.1 | 78.4 | 74.7 | 75.4 | 77.6 | 77.9 |

Figure 5. Adding Glance Fusion before Gaze Compression significantly improves performance across all datasets, with notable gains on fine-grained OCR tasks like TextVQA and VQAv2.

**Ablation study on $r$ and $p_r$.** In Gaze Compression, the initial compression layer index $r$ and the size of query embedding $p_r$ together shape the compression ratio. To evaluate the performance impact of each variable independently, we varied one while holding the other constant, facilitating a comparative analysis of their effects, as shown in Table 9. It reveals a direct proportional relationship between accuracy and the starting compression layer when $p_r$ is constant; larger $r$ corresponds to higher accuracy. This is logical, as premature compression may prevent the LLM from fully processing visual tokens. Similarly, opting for a larger query embedding size yields benefits, as it allows richer information capture during compression.

**Exploration on number of layers in Glance Fusion.** We fuse different numbers of ViT layers, with results displayed in Figure 6 (a). Interestingly, adding more ViT layers does not always result in proportional performance gains. This aligns with observations in Figure 1, suggesting significant redundancy across layers and certain intermediate layers in ViT not contributing to instruction comprehension. To balance performance across all datasets, we selected 4 layers for fusion in this study.

**Necessity of Instruction in Glance Fusion.** To demonstrate the significance of instruction in Glance Fuse, we compared it with straightforward feature averaging across layers. As illustrated in Figure 6 (b), indiscriminate fusion of ViT layers results in minimal improvement, whereas instruction-guided fusion significantly enhances features and boosts performance.

### A.3.4 More efficiency comparison

We provide comprehensive tables summarizing the training cost, inference latency, prefilling time, and throughput across different frameworks and compression configurations, offering a clear overview of the computational efficiency, as shown in Table 10.

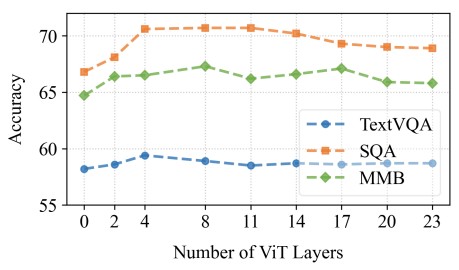

(a) Exploration on number of ViT layers

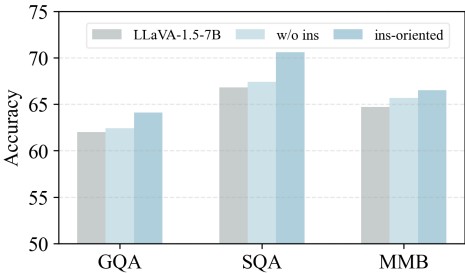

(b) Importance of Instruction-oriented integration

Figure 6: Ablation studies on Glance Fusion module.

Table 10: Comparison of computational efficiency between Glance2Gaze and baseline models. Metrics include training GPU hours, inference TFLOPs, latency, CUDA time, and throughput.

| Method | Training GPU Hours | Inference TFLOPs | Latency (ms) | CUDA Time (ms) | Throughput |
|---|---|---|---|---|---|
| *LLaVA-1.5-7B family* | | | | | |
| LLaVA-1.5-7B | 104 | 10.07 | 185.9 | 115 | 28.754 |
| Glance2Gaze (33% FLOPs) | 72 | 3.36 | 133.2 | 66 | 40.022 |
| Glance2Gaze (22% FLOPs) | 60 | 2.22 | 109.2 | 32 | 51.351 |
| Glance2Gaze (11% FLOPs) | 43 | 1.11 | 100.2 | 22 | 56.820 |
| *LLaVA-Next-7B family* | | | | | |
| LLaVA-Next-7B | 366 | 53.83 | 475.2 | 313 | 10.884 |
| Glance2Gaze (22% FLOPs) | 242 | 11.84 | 262.0 | 100 | 19.741 |
| Glance2Gaze (11% FLOPs) | 173 | 5.53 | 191.9 | 90 | 30.719 |
| Glance2Gaze (5% FLOPs) | 87 | 2.67 | 151.2 | 49 | 34.121 |
| *Video-LLaVA family* | | | | | |
| Video-LLaVA | 297 | 37.38 | 739.6 | 342.2 | 21.599 |
| Glance2Gaze (6% FLOPs) | 151 | 2.24 | 484.3 | 125.1 | 35.095 |

