# OpenReview forum: "Glance2Gaze: Efficient Vision-Language Models from Glance Fusion to Gaze Compression"
_NeurIPS.cc/2025/Conference — NeurIPS 2025 poster_

### Official Review · Reviewer_j7mC · 2025-06-18

**Clarity:** 3
**Significance:** 3
**Originality:** 3
**Rating:** 4
**Confidence:** 3

**Summary:**

This work points out that the existing approaches focus on reducing visual tokens at the visual encoder phase or the LLM decoder stage. However, this work suggests that there should be an initial global glance that precedes. This work has two key components: 1) Glance Fusion module: integrates a multi-layer vision transformer; 2) Gaze compression module: selectively compresses visual tokens based on the semantic relevance.

**Questions:**

N/A

**Ethical Concerns:**

["NO or VERY MINOR ethics concerns only"]

**Final Justification:**

There is no concerns.

**Limitations:**

Yes

**Quality:**

3

**Strengths And Weaknesses:**

Strength:
* The paper is well-written and clear presentation.
* The motivation is clear, presented in Figure 1. And the proposed method is based on the Motivation.
* The proposed method is clear and presented in Figure 2.
* The experiment supports the proposed methods.

Weakness:
* Provide the code in Appendix for reproduction.
* In Table 3, the Glance2Gaze  achieves better performance than naive Video-LLaVA on TGIF. However, the Galance2Gaze should achieve lower performance by employing compression.
* In Table 5, it seems that the effectiveness of the proposed fusion method gradually decrease, with the model size increase. What is the performance of the proposed fusion method with a larger model size?

---

> ### Author Rebuttal · Authors · 2025-07-30
>
> Thank you for your valuable feedback.
>
> Here we address the points you mentioned in the weaknesses part.
>
> 1. We appreciate the reviewer’s emphasis on reproducibility and the interest in accessing our code. Our code will be released upon paper’s acceptance.
>
> 2. We clarify that among the four video-QA datasets in Table 3 of the main text, only ActivityNet shows a slight performance gain for Glance2Gaze over naive Video-LLaVA. On TGIF and the other datasets, Glance2Gaze generally underperforms, aligning with the expectation that compression reduces available information and may impair performance. However, video data often contain significant redundancy and noise within visual tokens, which can hinder model comprehension. The compression applied by Glance2Gaze aims to distill salient visual information by removing redundant or noisy tokens. This simplification can yield a clearer, more focused representation, enabling better feature capture and, occasionally, improved performance on specific datasets or samples. Therefore, the occasional performance improvement post-compression is plausible and consistent with the data characteristics and model behavior.
>
> 3. Thanks for your insightful comment. We conducted additional experiments incorporating a larger model following Dense Connector for fair comparison. Specifically, we employed SigLIP-so-384px as the visual encoder alongside the Yi-34B large language model to evaluate the effectiveness of our Glance Fusion method at scale. Due to resource limitations, we utilized the LoRA fine-tuning technique for training the projector and the LLM, with lora_r and lora_alpha set to 128 and 256, respectively, consistent with Dense Connector’s settings. Given that SigLIP comprises 27 ViT layers, we applied Glance Fusion at layers [7, 14, 21, 27]. The results, summarized in the table below, indicate that Glance Fusion significantly enhances performance in larger models, surpassing Dense Connector.
>
>     | Method                   | training | GQA  | VQA v2 | SQA  | TextVQA | POPE |
>     |:------------------------:|:--------:|:----:|:------:|:----:|:-------:|:----:|
>     | baseline (SigLip-Yi-34B) | LoRA     |   62.4   |    81.1    |   78.8   |    64.7     |    81.4  |
>     | + Dense Connector        | LoRA     | 63.9 (+1.5) |    -    | 80.5 (+1.7) | 66.7 (+2.0)    |   80.5 (-0.9)  |
>     | + Glance Fusion          | LoRA     |   64.3 (+1.9)   |    83.5 (+2.4)    |   81.9 (+3.1)   |    67.5 (+2.8)     |    82.6 (+1.2)  |

---

> > ### Comment · Reviewer_j7mC · 2025-08-04
> >
> > After the above response, I will keep the score.

---

> > > ### Author Response · Authors · 2025-08-06
> > >
> > > Thank you for your thoughtful feedback and for considering our response. We appreciate your time and review.

---

### Official Review · Reviewer_CY4r · 2025-06-29

**Clarity:** 3
**Significance:** 3
**Originality:** 3
**Rating:** 5
**Confidence:** 4

**Summary:**

The paper introduces a new method to **compress visual tokens** in LLaVA-style multimodal LLMs (where a ViT vision encoder has its vision tokens projected to prefixed input tokens to the autoregressive LLM). The method "Glance2Gaze" consists of two modules:

First, a **Glance Fusion module** computes $\mathbf{V}_Q$ based on *both* vision tokens from multiple layers and text tokens from the instruction. The idea here is to extract information from multiple levels and based on the instruction, similar to the initial fixation stage in the human visual system.

Second, a **Gaze Compression** module replaces the full set of visual tokens with a subset that shrinks in progressive layers of the LLM transformer, to reach a certain target reduction in FLOPs. This set is computed via cross-attention using learned queries and $\mathbf{V}_Q$ from the Glance Fusion module as keys and values.

Glance2Gaze is compared to recent methods for visual compression in VLMs (FastV, SparseVLM, PDrop, VisionZip) on some common VLM and video benchmarks, reducing the FLOPs in the LLM to 33%/22%/11%, and shows a **consistent improvement**.

**Questions:**

1. In Formula (9) the symbol $\mathbf{V}^l_Q$ appears both on the left side and the right side. I think on the right side it should instead be $\mathbf{V}^{l-1}_Q$. I find it a bit confusing that there are "task-enhanced visual tokens" $\mathbf{V}_Q$ in Equation (8) and then there is a new symbol "visual tokens" $\mathbf{V}^l_Q$ that re-uses the same symbol but is different (the former being the integrated tokens from the Glance Fusion stage, and the latter being the consecutively compressed visual tokens from the Gaze Compression stage).

1. When computing the FLOPs it seems that only the LLM is taken into consideration? What fraction of the overall FLOPs are due to the vision encoder? Note that in Table 4 a 20x fold of FLOPs reduction is reported, but the speedup is 6.39x.

1. FLOPs computations in the appendix in Equation 2 and lines 20-27: Why would the visual token sequence compression FLOPs (first part of the formula) be included in "gaze compression"? I find that naming a bit confusing because the first part is simply the cost of the forward pass in the decoder. Still with the first part of the equation, why is $n$ constant throughout the decoder? I would have expected $n$ to be constant up to layer $r-1$, but then be replaced with $n_\text{ins} + p_l$ from the gaze compression, otherwise Gaze Compression would only add FLOPs to the sum (in the second part). As for the Glance Fusion FLOPs computation (appendix lines 18-19), it seems that the projections are missing from Equation 1.

1. In Figure 1, how are the tokens pruned? Are the residuals of pruned tokens still available to layers with a higher index?

1. In Figure 4, is it correct that the "Layer-specific Query" bars include the Glance Fusion stage and the only difference to the full Glance2Gaze is that the query embeddings are not shared?

Small comments

1. Line 95: "it resizes"
1. Line 108: missing space between the reference list and "Resampler".
1. Line 255: missing space between "Bind" and reference
1. Line 279: "but" is a bit misleading: one would usually expect the first part of the phrase to be positive (e.g. "reduces parameters") and the second part to be negative (e.g. "reduces performance") – but in this case both changes are negative
1. Line 637: where exactly is the training compute mentioned in the paper?

**Ethical Concerns:**

["NO or VERY MINOR ethics concerns only"]

**Final Justification:**

I maintain my rating of 5=accept. I think the paper presents an interesting method and shows a superior performance with previous sota in a fair comparison, so I find it valuable to share the work with the larger NeurIPS community.

As for the two weaknesses that I reported in my original review, I think the authors have addressed both of them satisfactorily:

1. The authors expanded on their FLOPs computations, and added a table that not only lists theoretical FLOPs, but also measures throughput and prefill latency, and compares their method to various baseline methods at different sparsification settings.

2. As for clarity, the authors have fixed some issues that I mentioned in my original review, and promised to improve the readability of some of the formulas that I found lacked a bit on clarity in the initial version of the manuscript.

**Limitations:**

yes

**Quality:**

3

**Strengths And Weaknesses:**

**Strengths**

1. The paper introduces an **innovative** biologically-inspired method for visual token compression in LaVA-style models. The Glance Fusion module leads to a set of vision tokens that is better suited for answering a question by incorporating the question in the selection process. The Gaze Compression mechanism still achieves good results at a high token compression level.

1. The method works with both images and videos and achieves a **consistent performance gain** over recent methods like VisionZip.

1. The paper is **well-written** and effectively illustrated, making the somewhat complex methodology easy to understand.

**Weaknesses**

1. The paper does not report the overall cost of training with Glance2Gaze. The **FLOPs computations** for inference are incomplete (e.g. the vision encoder is ignored), and there is only a single measurement of inference performance (single architecture, single FLOPs reduction setting).

1. **Clarity** of some of the equations and FLOPs computations **could be improved**, see details in section "Questions" below.

---

> ### Author Rebuttal · Authors · 2025-07-30
>
> Thank you for your thoughtful comments and suggestions.
>
> Here we address the points you mentioned in the weaknesses part.
>
> 1. We appreciate reviewer's attention on training cost, FLOPs computation and inference performance.
>
>     1). Here we provide comprehensive tables presenting training cost, inference latency, prefilling time, and throughput across various frameworks and compression settings to offer a clear picture of computational costs.
>
>     | Method                | Training GPU Hours | Inference TFLOPs | Latency (ms) | Cuda Time (ms) | Throughput |
>     |:---------------------:|:------------------:|:----------------:|:------------:|:--------------:|:----------:|
>     | LLaVA-1.5-7B         | 104                | 10.07            | 185.9        | 115            | 28.754     |
>     | Glance2Gaze (33% FLOPs) | 72                 | 3.36             | 133.2        | 66             | 40.022     |
>     | Glance2Gaze (22% FLOPs) | 60                 | 2.22             | 109.2        | 32             | 51.351     |
>     | Glance2Gaze (11% FLOPs) | 43                 | 1.11             | 100.2        | 22             | 56.82      |
>
>     | Method                | Training GPU Hours | Inference TFLOPs | Latency (ms) | Cuda Time (ms) | Throughput |
>     |:---------------------:|:------------------:|:----------------:|:------------:|:--------------:|:----------:|
>     | LLaVA-Next-7B         | 366                | 53.83            | 475.2        | 313            | 10.884     |
>     | Glance2Gaze (22% FLOPs) | 242               | 11.84            | 262.0        | 100            | 19.741     |
>     | Glance2Gaze (11% FLOPs) | 173               | 5.53             | 191.9        | 90             | 30.719     |
>     | Glance2Gaze (5% FLOPs)  | 87                | 2.67             | 151.2        | 49             | 34.121     |
>
>     | Method                | Training GPU Hours | Inference TFLOPs | Latency (ms) | Cuda Time (ms) | Throughput |
>     |:---------------------:|:------------------:|:----------------:|:------------:|:--------------:|:----------:|
>     | Video-LLaVA           | 297                | 37.38            | 739.6        | 342.2          | 21.599     |
>     | Glance2Gaze (6% FLOPs) | 151                 | 2.24             | 484.3        | 125.1          | 35.095     |
>
>     2). Following common practice in visual pruning literature (e.g., VisionZip, Pdrop, SparseVLM), we focus FLOPs measurement primarily on the computation related to visual tokens within the LLM. This is because pruning techniques selectively reduce computation in the LLM's visual token processing, while the visual encoder’s computational cost remains fixed and thus does not affect the pruning evaluation. Here we also present a detailed breakdown of FLOPs across different components of the vision-language model, including the visual encoder.
>
>      |         Glance2Gaze (11% FLOPs)          | in Visual Encoder | in Glance Fusion | in LLM  |
>     |:-----------------:|:-----------------:|:----------------:|:-------:|
>     | FLOPs             | 0.38 T             | 0.03 T           | 1.1 T   |
>     | proportion        | 0.2517             | 0.0199           | 0.7285  |
>
> 2. Thank you for the helpful suggestion. We will revise the relevant equations and FLOPs computations in the main text and supplementary to improve clarity and transparency.
>
> Then here are the answers to the Questions part.
>
> 1. Thank you for pointing this out. We acknowledge that the reuse of the same symbol in Equation (9) as well as the reuse from Equation (8) could lead to confusion regarding the semantics of the visual tokens at different stages of the pipeline. We will carefully revise the notation in the final paper.
>
> 2. Please see the answers to Weakness one for more comprehensive tables presenting training cost, inference latency, prefilling time, and throughput. Regarding the discrepancy between FLOPs reduction and wall-clock speedup, this stems from the well-established distinction between theoretical computational savings and practical runtime acceleration. FLOPs reductions do not translate linearly to real-world speedups due to factors such as memory access patterns and hardware constraints. Nonetheless, achieving a 6.39× speedup represents a substantial improvement in inference latency, underscoring the practical effectiveness of our approach.
>
> 3. We thank the reviewer for highlighting this point and apologize for any confusion caused in FLOPs computation of visual tokens.
>
>     1). Equation (2) aims to compute the total computational cost of processing visual tokens within the decoder, which consists of two parts:
>     -   the standard forward pass through the decoder (first term),
>     -   the additional cost introduced by the Gaze Compression module (second term).
>
>     We will revise the title to **“FLOPs within LLM (Visual Tokens)”**, to reflect that both standard forward and gaze compression contribute to the total cost.
>
>     2). Regarding the variable $n$ in the first term of Equation (2), your interpretation is correct. As clarified in Line 206 of the main text, $n$ denotes a variable that remains constant prior to layer $r$ and takes the value of $p_l$ thereafter.
>
>     3). We appreciate the reviewer pointing out the omission. The FLOPs computation for Glance Fusion should indeed include the cost of projection layers for both text and visual inputs. In Glance Fusion, the computational cost should include $S$ projection layers for text mapping and $S$ projection layers for visual mapping, which bring $SNd_vd_t$ and $SMd_t^2$ FLOPs respectively. After correction, the FLOPs for the Glance Fusion stage be 0.03T. Thus, the ratio of computation between Glance Fusion and the LLM is $0.03/1.1=0.0272$. We will update Equation (1) and the related text accordingly.
>
> 4. We compute visual token importance by averaging instruction-to-image attention scores extracted from the decoder's attention matrices. Specifically, we isolate the instruction-to-visual attention submatrix and calculate the mean attention each visual token receives from all instruction tokens, yielding a scalar importance score per token. Tokens are ranked accordingly, and those with the lowest scores are pruned based on a predefined ratio. Pruned tokens and their residuals are completely excluded from subsequent layers and computations, thus exerting no influence on downstream activations.
>
> 5. Your understanding is correct. The "Layer-specific Query" bars in Figure 4 do include the Glance Fusion stage. The only difference from the full Glance2Gaze model is that the query embeddings are not shared across layers.
>
> 6. Regarding the small comments, we thank the reviewer for pointing out grammatical and formatting errors, and we will address these in the final paper. Please refer to our response in Weakness one for clarification on training compute concerns.

---

> > ### Comment · Reviewer_CY4r · 2025-08-05
> >
> > I would like to thank the authors for their **concise and complete rebuttal** to all the points mentioned in my review.
> >
> > The tables would make a worthwhile addition for the appendix, as they give a good estimation of the real-world efficiency gains of the method (it might be interesting to contrast the same measurements with VisionZip).
> >
> > As for formulas (10) and (2, appendix), I do find the use of the symbol $n$ somewhat surprising, as it usually corresponds to a fixed quantity. Adding an index $n_l$ would be an easy way to clarify this point.

---

> ### Author Response · Authors · 2025-08-04
>
> Dear Reviewer,
>
> We sincerely appreciate your effort in reviewing. Could you kindly confirm if our rebuttal addresses your concerns, or if there are any further points we can clarify? We’d be happy to discuss.
>
> Best Regards,
>
> Author

---

> > ### Comment · Reviewer_CY4r · 2025-08-05
> >
> > Please excuse my late reply, but I had to focus on other papers with more borderline ratings over the weekend.

---

> > > ### Author Response · Authors · 2025-08-06
> > >
> > > Thank you for getting back to us, we completely understand the demands of the review process. We appreciate your time and thoughtful feedback on our paper.
> > >
> > >
> > > 1. We appreciate reviewer's attention on comparing Glance2Gaze with VisionZip, we provide the comparisons in below table. Our method provides a better tradeoff between speed and performance. We will include these runtime efficiency tables in the appendix to better illustrate the real-world efficiency gains.
> > >
> > >     | Method                | Inference TFLOPs | Latency (ms) | Cuda Time (ms) | Cuda Time / Token (ms) | Throughput |  Accuracy  Retention|
> > >     |:---------------------:|:----------------:|:------------:|:--------------:|:----------:|:----------:|:----------:|
> > >     | LLaVA-1.5-7B           | 10.07| 185.9  | 115 | 15.38 | 28.754 |   100%    |
> > >     | VisionZip (33% FLOPs)  | 3.32 | 139.2  | 76  | 15.23 | 32.51  |   99.1%   |
> > >     | Glance2Gaze (33% FLOPs)| 3.36 | 133.2  | 66  | 15.13 | 40.022 |   99.9%   |
> > >     | VisionZip (22% FLOPs)  | 2.22 | 119.6  | 50  | 15.15 | 40.03  |   97.9%   |
> > >     | Glance2Gaze (22% FLOPs)| 2.22 | 109.2  | 32  | 15.02 | 51.351 |   98.7%   |
> > >     | VisionZip (11% FLOPs)  | 1.11 | 108.7  | 39  | 15.03 | 44.72  |   94.9%   |
> > >     | Glance2Gaze (11% FLOPs)| 1.11 | 100.2  | 22  | 14.97 | 56.82  |   95.6%   |
> > >
> > >     | Method                | Inference TFLOPs | Latency (ms) | Cuda Time (ms) | Cuda Time / Token (ms) |Throughput |  Accuracy Retention |
> > >     |:---------------------:|:----------------:|:------------:|:--------------:|:----------:|:----------:|:----------:|
> > >     | LLaVA-Next-7B           | 53.83 | 475.2 | 313 | 24.02 | 10.884 |   100%   |
> > >     | VisionZip (22% FLOPs)   | 11.84 | 293.7 | 112 | 23.97 | 17.47  |   97.9%  |
> > >     | Glance2Gaze (22% FLOPs) | 11.84 | 262.0 | 100 | 23.78 | 19.741 |   99.2%  |
> > >     | VisionZip (11% FLOPs)   | 5.92  | 204.3 | 103 | 23.82 | 22.93  |   96.3%  |
> > >     | Glance2Gaze (11% FLOPs) | 5.53  | 191.9 | 90  | 23.66 | 30.719 |   97.6%  |
> > >     | VisionZip (5% FLOPs)    | 2.76  | 192   | 57  | 23.67 | 26.983 |   93.6%  |
> > >     | Glance2Gaze (5% FLOPs)  | 2.67  | 151.2 | 49  | 23.54 | 34.121 |   94.8%  |
> > >
> > >     | Method                | Inference TFLOPs | Latency (ms) | Cuda Time (ms) | Cuda Time / Token (ms) |Throughput |  Accuracy  Retention|
> > >     |:---------------------:|:----------------:|:------------:|:--------------:|:----------:|:----------:|:----------:|
> > >     | Video-LLaVA            | 37.38     |    739.6   |    342.2   |  24.96 |   21.599  |    100%   |
> > >     | VisionZip (6% FLOPs)   | 2.24      |    513.9   |    133.8   |  24.79 |   33.139  |    93.2%  |
> > >     | Glance2Gaze (6% FLOPs) | 2.24      |    484.3   |    125.1   |  24.46 |   35.095  |    96.2%  |
> > >
> > > 2. We thank the reviewer for pointing out the potential ambiguity in our use of the symbol in Formulas (10) and (2) in the appendix. We will revise the notation accordingly in the final paper.

---

### Official Review · Reviewer_TFE2 · 2025-07-01

**Clarity:** 3
**Significance:** 2
**Originality:** 2
**Rating:** 4
**Confidence:** 5

**Summary:**

This paper focuses on the efficiency of MLLMs, with one idea that to observe the image information, it is necessary to first conduct a preliminary global browsing, and then focus on semantically prominent regions. A two-stage framework, namely Glance2Gaze, is proposed. The glance fusion module integrates multi-layer vision transformer features with text-aware attention to generate semantically global representations, and the gaze compression module utilizes a query-guided mechanism to selectively compress visual tokens by semantic relevance. Experiments are conducted in a series of VQA benchmarks.

**Questions:**

[-] Some Surprising Results. In Table 1, on the benchmarks of SQA, MMVet, and SEED, Glance2Gaze even outperforms LLaVA-1.5-7B, which means that compressing tokens for efficiency actually enhances performance. Please provide more discussions to explain these phenomena.

**Ethical Concerns:**

["NO or VERY MINOR ethics concerns only"]

**Final Justification:**

Firstly, I would like to thank the author for their rebuttal. During this period, the author provides detailed responses to my key concerns, which further enhances the quality of the method. Taking into account both innovation and practicality, I have decided to slightly increase my rating to borderline acceptance.

**Limitations:**

Yes

**Quality:**

3

**Strengths And Weaknesses:**

[+] The manuscript is well written, with clear logics and sufficient formulations.

[+] Efficient MLLMs have great practicality for the community, thus this work is on the right path.

[-] Many bad cases from the cognitive perspective. For some simple scenes, from glance to gaze is feasible. While in many complex scenes, a single image has multiple focuses, and sometimes, these focuses may overlap. Such complex scenes may lead to failure of Glance2Gaze.

[-] Unclear Efficiency. In Sec 3.3, this paper shows some cases of LLaVA-1.5-7B in terms of FLOPs to prove the efficiency. However, this is not comprehensive and requires more evaluations, such as lantency and throughput. Please report more quantitative results.

[-] How to deal with image-text unpatching? As one generalized model, MLLM also encounters many special inputs, e.g., the text inputs are unrelated to the image inputs. In this case, glance fusion will be misled, causing gaze compression to fail.

[-] Insufficient Ablations. How to prove the effectiveness of the glance fusion module?  For gaze compression, will learnable queries be better than directly calculating the similarity of image-text?

---

> ### Author Rebuttal · Authors · 2025-07-30
>
> Thank you for taking the time to review our paper.
>
> Here we address the points you mentioned in the weaknesses part.
>
> 1. Thank you for the insightful comment. We agree that cognitively complex scenes, particularly those with multiple and potentially overlapping focal points, pose challenges for attention-reduction mechanisms. To explore this, we evaluated Glance2Gaze on counting tasks from the VQA v2 dataset, which exemplify such complexity due to scattered and partially overlapping target objects requiring distributed attention. We compare with LLaVA-1.5-7B under various reduction ratios. As shown in the table below, Glance2Gaze consistently preserves 93.43%–99.20% of its full-retention accuracy, significantly outperforming VisionZip. These results suggest that Glance2Gaze does not concentrate on a single region, but instead maintains broad coverage of all relevant areas in complex visual scenes. We welcome further discussion if you were referring to other types of complex scenes.
>
>     | Method      | FLOPs | Acc          |
>     |:-----------|:-----|:------------|
>     | LLaVA-1.5-7B| 100%  | 62.4 (100%)  |
>     | VisionZip   | 33%   | 61.8 (99.04%)|
>     | Glance2Gaze | 33%   | 61.9 (99.20%)|
>     | VisionZip   | 22%   | 59.3 (95.03%)|
>     | Glance2Gaze | 22%   | 60.4 (96.79%)|
>     | VisionZip   | 11%   | 57.0 (91.35%)|
>     | Glance2Gaze | 11%   | 58.3 (93.43%)|
>
> 2. In Section 3.3 of the main paper, we report FLOPs and also include CUDA prefilling time in Table 4 as a proxy for latency. We now included a comparison of overall latency and throughout in below table on the TextVQA dataset. Glance2Gaze achieves a **3.15× speedup in overall inference latency**, a **6.39× speedup in prefilling time** and a **3.135× increase in throughput** compared to the vanilla model. These results confirm that our method offers substantial end-to-end efficiency gains, not just in the prefilling phase.
>
>     | Method      | TFLOPs | Cuda Time (ms) | Latency (ms) | Throughput |
>     |:-----------:|:------:|:--------------:|:------------:|:------------:|
>     | LLaVA-NeXT  | 53.83  |      313       |     475      |    10.884 | |
>     | FastV       | 2.76   | 170 (1.84 $\times$) | 358 (1.33 $\times$) | 12.272 (1.128 $\times$) |
>     | SparseVLM   | 2.76   | 183 (1.71 $\times$) | 379 (1.25 $\times$) | 13.233 (1.216 $\times$) |
>     | VisionZip   | 2.76   |  57 (5.49 $\times$) | 192 (2.47 $\times$) | 26.983 (2.479 $\times$) |
>     | Glance2Gaze | 2.67   |  49 (6.39 $\times$) | 151 (3.15 $\times$) | 34.121 (3.135 $\times$) |
>
> 3. Thank you for raising this point. While most standard VQA datasets contain well-aligned image-text pairs, we recognize the need for benchmarks that intentionally introduce image-text mismatch while still requiring visual grounding. Despite an extensive search, we were unable to identify publicly available datasets with these characteristics, and we would welcome recommendations for future study. As an alternative, we evaluated POPE, a benchmark for hallucination scenarios where textual descriptions reference non-existent objects. Results in Tables 1~2 in the main paper demonstrate Glance2Gaze's robustness across reduction ratios, significantly outperforming other pruning methods.
>
> 4. We appreciate reviewer's attention on effectiveness of Glance Fusion module and learnable queries in Gaze Compression.
>
>     1). In the main paper, we conducted targeted ablations to evaluate the contribution of the Glance Fusion module. As shown in Figure 4, the “Gaze Compression Only” baseline disables Glance Fusion, revealing a notable performance drop discussed in lines 282–288. Additionally, Table 5 compares Glance Fusion as a plug-and-play component against alternative fusion strategies, with detailed analysis in lines 289–296. Both results consistently confirms Glance Fusion's effectiveness.
>
>     2). Accordingly, we implemented a baseline that removes tokens solely based on instruction-to-image attention scores starting from the $r$-th decoder layer. To ensure fairness, we fixed $r=9$ and $p_r=256$, replicating the exponential token retention schedule used in Glance2Gaze. At each layer from $r$ onward, visual tokens were ranked by their attention importance, and the lowest-ranked tokens were pruned according to the predetermined count. As shown in the table below, this attention-based pruning causes substantial performance degradation. Even after fine-tuning (The visual encoder is frozen while all other parameters are fine-tuned for 1 epoch, using identical training configurations as Glance2Gaze), the baseline fails to match Glance2Gaze’s performance, confirming that static pruning based on local attention scores lacks the adaptability required for task-specific compression.
>
>     | Method   | POPE | SQA  | MME  | GQA  | SEED | MMB  | VQA v2 | TextVQA | MMVet | LLaVA-B | Avg.|
>     |:--------------:|:----:|:----:|:----:|:----:|:----:|:----:|:------:|:-------:|:-----:|:-------:|:-------:|
>     | w/o finetune     | 83.4 | 69.1 | 1775 | 59.7 | 54.7 | 63.0 | 74.6   | 55.8    | 30.6  | 62.0    | 96.1% |
>     | w finetune     | 84.7 | 69.1 | 1793 | 60.8 | 56.8 | 63.2 | 75.7   | 56.4    | 31.4  | 65.2    |97.9% |
>     | Glance2Gaze    | 85.5 | 70.4 | 1812 | 61.5 | 58.7 | 64.5 | 77.6   | 57.2    | 32.7  | 66.4    |99.9% |
>
> Then here are the answers to the Questions part.
>
> 1. While full retention theoretically preserves maximal information, it may introduce redundant or irrelevant features that degrade model performance, an effect similarly observed in early deep networks, where increasing depth did not always yield better results compared to shallower architectures. In contrast, our approach avoids naive compression by employing a structured fusion mechanism (e.g., Glance Fusion) that selectively integrates visual signals aligned with task-relevant semantics.

---

> ### Author Response · Authors · 2025-08-04
>
> Dear Reviewer,
>
> We sincerely appreciate your effort in reviewing. Could you kindly confirm if our rebuttal addresses your concerns, or if there are any further points we can clarify? We’d be happy to discuss.
>
> Best Regards,
>
> Author

---

### Official Review · Reviewer_kUtD · 2025-07-07

**Clarity:** 4
**Significance:** 3
**Originality:** 4
**Rating:** 4
**Confidence:** 4

**Summary:**

This paper introduces Glance2Gaze, a cognitively-inspired framework for efficient vision-language models that mimics human eye’s movement patterns. The framework consists of two components: Glance Fusion, which integrates multi-layer ViT features with text-aware attention, and Gaze Compression, which progressively reduces visual tokens inside of LLMs using a shared query pool. It achieves good performance while pruning the input tokens that could significantly reduce computation cost.

**Questions:**

(1) How were the specific layer indices chosen for Glance Fusion? Were other layer indices tested for Glance Fusion?
(2) Can you provide a breakdown of the computational overhead with the mentioned "0.17%" in section 3.3? For the computation cost of the language model, does it differ between prefill and inference?

**Ethical Concerns:**

["NO or VERY MINOR ethics concerns only"]

**Limitations:**

Here’s one suggestion to improve the insight of the paper. The idea of Gaze Compression is impressive. Similar to table 5, adding the comparison of Gaze Compression to other methods would further improve the necessity, e.g. comparing it with token pruning using only attention score with the same compression ratio in each layer.

**Quality:**

3

**Strengths And Weaknesses:**

Paper Strength
(1) Extensive evaluation on 10 image understanding and 4 video understanding benchmarks demonstrates consistent improvements, with particularly impressive results maintaining good performance at 33%,22% and 11% FLOPs.

(2)The paper includes reasonable ablation studies, visualizations of the compression process, and detailed efficiency comparisons including both FLOPs and prefill latency measurements.

(3)The text-aware attention mechanism in Glance Fusion and the shared query pool design in Gaze Compression are novel contributions that effectively balance efficiency and performance.



Paper Weakness
(1) The choice of ViT layers for Glance Fusion and compression sequence for Gaze Compression seems arbitrary. For Glance Fusion, it lacks discussion why choosing selected layers L={7,13,9,23} to enhance semantic information is necessary. Similarly, the paper uses exponential decay (e.g., [256, 128, 64, ..., 2]) for compression but doesn't provide in-depth analysis.

(2) Section 4.2 claims Glances2Gaze achieves inference speedup up to 6.39x compared to vanilla method, whereas table 4 only provides latency comparison of prefilling time. The overall inference time comparison is untested.

(3) Training of Glance2Gaze requires finetuning the whole language model, compared to the training-free method such as VisionZip, it introduces additional training costs despite performance gain.

(4) While the cognitive inspiration is compelling, the paper lacks rigorous theoretical analysis explaining why Glance2Gaze could outperform previous work such as VisionZip.

---

> ### Author Rebuttal · Authors · 2025-07-30
>
> Thank you for your valuable feedback.
>
> Here we address the points you mentioned in the weaknesses part.
>
>  1. We appreciate the reviewer’s attention to the choice of ViT layers and compression sequence.
>
>     1).  Our selection of layers {7,13,19,23} was guided by both visual insights and empirical evidence. As illustrated in Figure 1 (left) in the main paper, while ViT layers exhibit varied responses to instruction tokens, adjacent layers yield highly correlated representations. For empirical evidence, we computed average attentional responses of text tokens across 24 ViT layers using the TextVQA and ScienceQA datasets, resulting in the following normalized distribution:
>
>     | Layer Range | Values
>     |-------------|------------------------------------------------------------------------|
>     | 1–7         | 0.0393, 0.0393, 0.0395, 0.0403, 0.0405, 0.0416, 0.0442                  |
>     | 8–17        | 0.0400, 0.0401, 0.0398, 0.0398, 0.0397, 0.0408, 0.0394, 0.0392, 0.0395, 0.0395 |
>     | 18–20       | 0.0406, 0.0432, 0.0412                                                 |
>     | 21–23       | 0.0627, 0.0634, 0.0658                                                 |
>
>     Within each range, we selected the layer exhibiting the highest response correlation to instruction tokens, yielding the set {7,13,19,23}. For further ablation, we tested the second-best layers in each range, {6,9,20,22}. As shown in the table below, both sets significantly outperformed the baseline, but the original selection gave marginally higher gains.
>
>     | Method                | GQA         | VQA v2      | SQA         | TextVQA     | POPE        |
>     |:-----------------------|:-------------|:-------------|:-------------|-------------|:-------------|
>     | LLaVA-1.5-7B          | 62.0        | 78.5        | 66.8        | 58.2        | 85.9        |
>     | + Glance Fusion \([7, 13, 19, 23]\) | 64.1 (+2.1) | 79.6 (+1.1) | 70.6 (+3.8) | 59.4 (+1.2) | 87.2 (+1.3) |
>     | + Glance Fusion \([6, 9, 20, 22]\)  | 64.1 (+2.1) | 79.4 (+0.9) | 70.5 (+3.7) | 59.4 (+1.2) | 86.9 (+1.0) |
>
>     2). An exponential decay schedule (the function is ${\frac 2 {p_r}}^{1/(R-r)}$, specific number for each layer are given in Table 1 in the supplementary) was adopted in Gaze Compression to enable stable and progressive compression of visual information. We here additionally evaluated two variant schedules: fast-to-slow decay and slow-to-fast decay. Both resulted in performance degradation, with the fast-to-slow strategy yielding the most significant drop. We hypothesize that aggressive early compression leads to substantial information loss, causing irreversible harm. Results are presented in the table below.
>
>     | Strategy               | POPE  | SQA   | MME  | GQA  | SEED | MMB  | VQAv2 | TextVQA | MMVet | LLaVA-B | Avg. |
>     |:-----------------------|:-----:|:-----:|:----:|:----:|:----:|:----:|:-----:|:-------:|:-----:|:-------:| :-------:|
>     | Exponential Decay (Adopted) | 83.1  | 69.1  | 1722 | 56.9 | 53.9 | 61.7 | 74.9  | 55.7   | 31.4  | 64.1   | 95.6%|
>     | Fast->Slow             | 82.7  | 68.4  | 1686 | 55.4 | 53.8 | 59.7 | 72.6  | 54.8   | 30.7  | 63.3   | 93.9%|
>     | Slow->Fast             | 83.0  | 69.2  | 1712 | 56.0 | 53.2 | 61.2 | 73.8  | 55.2   | 31.3  | 63.7  | 94.9% |
>
> 2. Thank you for pointing this out. We now included a comparison of overall latency in below table on the TextVQA dataset. Glance2Gaze achieves a **3.15× speedup in overall inference latency** and a **6.39× speedup in prefilling time** compared to the vanilla model. These results confirm that our method offers substantial end-to-end efficiency gains, not just in the prefilling phase.
>
>     | Method      | TFLOPs | Cuda Time (ms) | Latency (ms) |
>     |:-----------:|:------:|:--------------:|:------------:|
>     | LLaVA-NeXT  | 53.83  |      313       |     475      |
>     | FastV       | 2.76   | 170 (1.84 $\times$) | 358 (1.33 $\times$) |
>     | SparseVLM   | 2.76   | 183 (1.71 $\times$) | 379 (1.25 $\times$) |
>     | VisionZip   | 2.76   |  57 (5.49 $\times$) | 192 (2.47 $\times$) |
>     | Glance2Gaze | 2.67   |  49 (6.39 $\times$) | 151 (3.15 $\times$) |
>
> 3. We acknowledge that Glance2Gaze requires model finetuning, unlike training-free approaches. However, the results reported for VisionZip correspond to its fine-tuned variant (as noted in their paper), ensuring a fair comparison. Importantly, despite requiring training, Glance2Gaze significantly reduces overall training cost: as shown in our response to Reviewer CY4r (W1), it achieves a **76.2% reduction in LLaVA-Next-7B training time**, operating at only **5% of full-model FLOPs**, while retaining **94.8%** of the original model’s performance. This exceeds VisionZip’s fine-tuned performance (**93.6%**) under comparable compression, demonstrating that the modest training overhead yields superior efficiency–performance trade-offs and positions Glance2Gaze as a highly competitive alternative.
>
> 4. We thank the reviewer for highlighting the cognitive motivation behind Glance2Gaze. Conceptually, Glance2Gaze is motivated by a well-established cognitive principle, it explicitly separates what to attend to from how to resolve it. This two-stage abstraction enhances spatial reasoning and compositional generalization, contributing to improved performance. Our current goal is to  demonstrate the proposed approach's effectiveness through extensive experiments. We acknowledge that a stronger theoretical foundation, e.g., via information-theoretic or attention-based modeling, would further strengthen the work and we consider it an important direction for future research.
>
> Then here are the answers to the Questions part.
>
>  1. Please see the answers to Weakness one.
>  2. We thank the reviewer for raising this point.
>
>     1). As mentioned in lines 200–201 of the main paper, the breakdown is provided in Supplementary A.2. However, we would like to acknowledge that Equation (1) in the Glance Fusion section of the supplementary initially omitted the projection operations. After incorporating all relevant projection layers, we recalculated the overhead introduced by Glance Fusion to be approximately 0.03 TFLOPs. Given that the language model requires around 1.1 TFLOPs to process visual tokens during the prefilling stage, the updated overhead ratio is 0.0272. We will revise both the supplementary and main text to reflect this corrected value for transparency and accuracy.
>
>     2). The primary distinction between the prefilling and decoding (inference) stages of LLMs lies in token processing. During prefilling, all input tokens, including instruction text and image tokens, are processed in parallel. In contrast, decoding omits image token computation, as their hidden states are cached in the KV store. Consequently, the FLOPs associated with image tokens pertain exclusively to the prefilling stage.
>
> Then here are the answers to the Limitations part.
>
> 1. We appreciate the reviewer’s valuable suggestion to compare Gaze Compression with attention-based token pruning at a fixed compression ratio. Accordingly, we implemented a baseline that removes tokens solely based on instruction-to-image attention scores starting from the $r$-th decoder layer. To ensure fairness, we fixed $r=9$ and $p_r=256$, replicating the exponential token retention schedule used in Glance2Gaze. At each layer from $r$ onward, visual tokens were ranked by their attention importance, and the lowest-ranked tokens were pruned according to the predetermined count. As shown in the table below, this attention-based pruning causes substantial performance degradation. Even after fine-tuning (The visual encoder is frozen while all other parameters are fine-tuned for 1 epoch, using identical training configurations as Glance2Gaze), the baseline fails to match Glance2Gaze’s performance, confirming that static pruning based on local attention scores lacks the adaptability required for task-specific compression.
>
>     | Method   | POPE | SQA  | MME  | GQA  | SEED | MMB  | VQA v2 | TextVQA | MMVet | LLaVA-B | Avg.|
>     |:--------------:|:----:|:----:|:----:|:----:|:----:|:----:|:------:|:-------:|:-----:|:-------:|:-------:|
>     | w/o finetune     | 83.4 | 69.1 | 1775 | 59.7 | 54.7 | 63.0 | 74.6   | 55.8    | 30.6  | 62.0    | 96.1% |
>     | w finetune     | 84.7 | 69.1 | 1793 | 60.8 | 56.8 | 63.2 | 75.7   | 56.4    | 31.4  | 65.2    |97.9% |
>     | Glance2Gaze    | 85.5 | 70.4 | 1812 | 61.5 | 58.7 | 64.5 | 77.6   | 57.2    | 32.7  | 66.4    |99.9% |

---

> > ### Comment · Reviewer_CY4r · 2025-08-05
> >
> > Short question about **inference latency** measurement: What exactly is measured in the second column of the added table? Does "151 ms latency" mean the latency until the first token (or complete answer) is generated or is it the latency per token? I would find the "Cuda time / token" more useful because that would not include the prefill time that is already reported in the first column.
> >
> > As for the question about the computational overhead in prefill vs. inference, it might be useful to look at memory bandwidth in addition to FLOPs (because fewer vision tokens and consequently a smaller KV cache would lead to faster inference with constant memory bandwidth).

---

> > > ### Author Response · Authors · 2025-08-06
> > >
> > > Thanks to your valuable suggestions. Here are answers to your questions.
> > >
> > > 1. In the table, 'Latency (ms)' reflects the time taken to generate the complete response, incorporating the prefilling time outlined in the first column. Following your suggestion, we provide 'CUDA time per token' in the subsequent table for a more granular analysis, representing the decoding duration for each text token generated, which excludes the prefilling time. While Glance2Gaze is primarily designed to reduce prefill latency, it also achieves competitive decoding efficiency, exhibiting slightly lower per-token CUDA time than alternative methods. Moreover, it retains 94.8% of the original accuracy, offering a superior trade-off between speed and performance.
> > >
> > >     | Method      | TFLOPs | Cuda Time-Prefilling (ms) | Cuda Time-Decoding (ms/Token) |  Overall latency (ms) |Accuracy Retention |
> > >     |:-----------:|:------:|:--------------:|:------------:|:------------:|:------------:|
> > >     | LLaVA-NeXT  | 53.83  |      313       |     24.02     |     475      |    100%   |
> > >     | FastV       | 2.76   | 170 (1.84 $\times$) |   23.88 (1.006 $\times$)   |    358 (1.33 $\times$)  |   -      |
> > >     | SparseVLM   | 2.76   | 183 (1.71 $\times$) |   23.97 (1.002 $\times$)   |    379 (1.25 $\times$)  |   85.0%  |
> > >     | VisionZip   | 2.76   |  57 (5.49 $\times$) |   23.67 (1.015 $\times$)   |    192 (2.47 $\times$)  |   93.6%  |
> > >     | Glance2Gaze | 2.67   |  49 (6.39 $\times$) |   23.54 (1.020 $\times$)   |    151 (3.15 $\times$)  |   94.8%  |
> > >
> > > 2. We thank the reviewer for highlighting the importance of memory bandwidth beyond FLOPs. Reducing visual tokens during prefilling decreases the vision KV cache size, thereby lowering memory pressure and accelerating inference. The calculated KV cache sizes for the vision component are presented in the table below. Glance2Gaze uses only 4.7% of the KV cache relative to the full model—less than VisionZip’s 5.6%—while achieving higher accuracy (94.8% vs. 93.6%). Considering both KV cache size and real-time performance, our method offers genuine acceleration in prefilling and overall inference, with superior accuracy retention compared to other approaches.
> > >
> > >     | Method      | TFLOPs | KV Cache (GB)       |  Accuracy  Retention|
> > >     |:-----------:|:------:|:--------------:|:------------:|
> > >     | LLaVA-NeXT  | 53.83  |  1.510            |   100%       |
> > >     | VisionZip   | 2.76   |  0.084 (5.6%)     |   93.6%      |
> > >     | Glance2Gaze | 2.67   |  0.071 (4.7%)     |   94.8%      |

---

> ### Author Response · Authors · 2025-08-04
>
> Dear Reviewer,
>
> We sincerely appreciate your effort in reviewing. Could you kindly confirm if our rebuttal addresses your concerns, or if there are any further points we can clarify? We’d be happy to discuss.
>
> Best Regards,
>
> Author

---

### Note · Authors · 2025-08-12

We appreciate the reviewers’ constructive feedback. This paper presents a significant advancement in efficient multimodal large language models (MLLMs) via the biologically inspired **Glance2Gaze** framework, which integrates global multi-layer visual fusion (“Glance Fusion”) with query-guided progressive token pruning (“Gaze Compression”). Our approach achieves consistent performance improvements alongside various computational savings across diverse VQA benchmarks and video datasets.

**Key strengths highlighted by reviewers include:**

- The clear motivation, solid technical foundation, and impressive empirical results with comprehensive comparative analysis (kUtD, CY4r, j7mC).
- The practical importance and applicability of our approach in improving efficiency for MLLMs (TFE2, CY4r).
- The manuscript’s clarity and logical presentation across all reviews.

**In response to reviewer concerns, we provided the following clarifications and additional results:**

- **Efficiency Evaluation:** We expanded our analysis to include training GPU hours, inference time, prefilling time, throughput, and FLOPs across multiple compression ratios, benchmarking against the baseline and VisionZip (kUtD, TFE2, CY4r). This demonstrates that Glance2Gaze achieves superior speedups and accuracy retention in practice.
- **Ablation Studies:** We conducted further experiments on hyperparameter choices, scalability to larger models, and comparisons with token pruning baselines, confirming the robustness and generalizability of our approach (kUtD, j7mC, TFE2).
- **Complex  Scenarios:** Additional discussions on tasks involving multiple visual focus (e.g., counting) and image-text mismatches (hallucination scenarios) validate the method’s resilience in challenging settings (TFE2).
- **Fine-tuning Clarification:** We clarified that VisionZip results are based on a fine-tuned variant, ensuring fair comparisons (kUtD).
- **Theoretical Analysis:** While our current work focuses on empirical validation, we acknowledge the potential to strengthen the theoretical foundation, which we identify as an important future direction (kUtD).

The reviewers’ insightful feedback mainly regarding experimental details, particularly efficiency evaluations and robustness in complex scenarios. We have carefully responded with additional experiments and clarifications, further enhancing the rigor and completeness of our work.

---

### Decision · Program_Chairs · 2025-09-17

**Decision:**

Accept (poster)

**Comment:**

This paper makes a significant contribution to the important and practical problem of VLM efficiency. The proposed Glance2Gaze framework is novel, well-motivated, and empirically powerful. The paper received 3 borderline accepts and an accept. Authors have addressed all major reviewer concerns. Therefore, we have decided to accept the paper.